# Continuous Monitoring of Vital Signs Using Cameras: A Systematic Review

**DOI:** 10.3390/s22114097

**Published:** 2022-05-28

**Authors:** Vinothini Selvaraju, Nicolai Spicher, Ju Wang, Nagarajan Ganapathy, Joana M. Warnecke, Steffen Leonhardt, Ramakrishnan Swaminathan, Thomas M. Deserno

**Affiliations:** 1Peter L. Reichertz Institute for Medical Informatics of TU Braunschweig and Hannover Medical School, D-38106 Braunschweig, Germany; vinothini.selvaraju@plri.de (V.S.); nicolai.spicher@plri.de (N.S.); ju.wang@plri.de (J.W.); nagarajan.ganapathy@plri.de (N.G.); joana.warnecke@plri.de (J.M.W.); 2Non-Invasive Imaging and Diagnostic Laboratory, Biomedical Engineering, Department of Applied Mechanics, Indian Institute of Technology Madras, Chennai 600036, India; sramki@iitm.ac.in; 3Chair for Medical Information Technology, Helmholtz-Institute for Biomedical Engineering, RWTH Aachen University, D-52074 Aachen, Germany; leonhardt@hia.rwth-aachen.de

**Keywords:** camera, vital sign, heart rate, respiratory rate, body temperature, oxygen saturation, blood pressure, noncontact, contactless, continuous monitoring, remote health care

## Abstract

In recent years, noncontact measurements of vital signs using cameras received a great amount of interest. However, some questions are unanswered: (i) Which vital sign is monitored using what type of camera? (ii) What is the performance and which factors affect it? (iii) Which health issues are addressed by camera-based techniques? Following the preferred reporting items for systematic reviews and meta-analyses (PRISMA) statement, we conduct a systematic review of continuous camera-based vital sign monitoring using Scopus, PubMed, and the Association for Computing Machinery (ACM) databases. We consider articles that were published between January 2018 and April 2021 in the English language. We include five vital signs: heart rate (HR), respiratory rate (RR), blood pressure (BP), body skin temperature (BST), and oxygen saturation (SpO_2_). In total, we retrieve 905 articles and screened them regarding title, abstract, and full text. One hundred and four articles remained: 60, 20, 6, 2, and 1 of the articles focus on HR, RR, BP, BST, and SpO_2_, respectively, and 15 on multiple vital signs. HR and RR can be measured using red, green, and blue (RGB) and near-infrared (NIR) as well as far-infrared (FIR) cameras. So far, BP and SpO_2_ are monitored with RGB cameras only, whereas BST is derived from FIR cameras only. Under ideal conditions, the root mean squared error is around 2.60 bpm, 2.22 cpm, 6.91 mm Hg, 4.88 mm Hg, and 0.86 °C for HR, RR, systolic BP, diastolic BP, and BST, respectively. The estimated error for SpO_2_ is less than 1%, but it increases with movements of the subject and the camera-subject distance. Camera-based remote monitoring mainly explores intensive care, post-anaesthesia care, and sleep monitoring, but also explores special diseases such as heart failure. The monitored targets are newborn and pediatric patients, geriatric patients, athletes (e.g., exercising, cycling), and vehicle drivers. Camera-based techniques monitor HR, RR, and BST in static conditions within acceptable ranges for certain applications. The research gaps are large and heterogeneous populations, real-time scenarios, moving subjects, and accuracy of BP and SpO_2_ monitoring.

## 1. Introduction

Physiological processes of the human body are associated with electrical, chemical, kinematic, and acoustic changes [1]. Although their measurement may not provide direct physiological evaluation or diagnosis, they yield valuable information regarding human activities and physiology [2]. This information is particularly useful in monitoring the cardiovascular system [3]. Several authors such as Lu et al., have comprehensively investigated electrophysiological signals such as electrocardiography (ECG) or optical signals such as photoplethysmography (PPG) to record vital signs [4]. Commonly, the four primary vital signs are heart rate (HR), respiratory rate (RR), blood pressure (BP), and body temperature (BT), and oxygen saturation (SpO_2_) is usually referred to as the fifth vital sign [5,6].

Vital signs offer information on the cardiovascular and respiratory systems [7]. The HR measures the number of heart contractions per minute and is usually given in beats per minute (bpm). The respiratory cycle includes inhalation and exhalation, and the RR is typically measured in cycles per minute (cpm) [8]. The maximum pressure exerted in the arteries during cardiac contraction is known as systolic BP, whereas the minimum pressure during cardiac relaxation is described as diastolic BP [9]. BT is a measurement of the body’s capacity to produce and release heat and is an early indicator of illness [10]. SpO_2_ measures the relative concentration of oxygenated hemoglobin with respect to the total amount of hemoglobin, and it is an essential physiological parameter to assess the oxygen supply to the human body [11,12]. These vital signs are monitored routinely in clinical practice using contact-based technology such as ECG recorders, respiratory belts, finger pulse oximeters, sphygmomanometers, and thermometers [5,13,14]. However, contact-based measures are obtrusive and may result in skin irritations. For instance, wet ECG electrodes need replacement after 48 h. Therefore, noncontact and unobtrusive vital sign monitoring is an emerging field of research [14]. For example, capacitive coupled ECG can be used to measure the electrical activity of the heart in an indirect way. It is recorded without physical contact to the skin through clothing and can be integrated into chairs, beds, or driver seats [15]. However, capacitive ECG is strongly affected by movement artifacts, coupling impedance fluctuation, and electromagnetic interference [16]. On the other hand, camera-based techniques are the focus of current research [13,17]. 

Image recording systems are inexpensive, broadly available, and easily combined with the increasing processing power of microcomputer systems. Utilizing a camera, Wu et al., firstly obtained dermal perfusion changes from the skin surface [18]. The underlying idea of image recording is similar to that of detecting the amplitude of finger pulse. Due to the contraction and relaxation phases of the human heart, the blood travels throughout the vascular system. It can be detected in every part of the body to different degrees. These pulsatile blood flow causes subtle changes in the skin color that is detected by image recording devices [18,19,20]. It is also referred as the color-based approach. Another approach is to measure the cyclic motions of the body caused by cardiorespiratory activity. It is known as the motion-based approach or ballistocardiography (BCG) [21,22]. The use of image or video data might offer noninvasive and unobtrusive measurement of vital signs [7,13]. 

Camera-based monitoring is applicable in clinical and nonclinical settings. In clinical settings, target groups include geriatric patients [23], newborns [24,25,26,27,28], patients in intensive care units (ICU) [29], patients undergoing magnetic resonance imaging [30], or patients before, during, and after surgery [31]. Noncontact measurement reduces the risk of infection and increases the patient comfort [23] and does not disturb the patient, which is important not only for detecting sleep disorders (e.g., apnea) but also for hypertension, heart failure, and arrhythmia [32,33,34]. Nonclinical applications are in-home [33] or in-vehicle monitoring (e.g., driver) [35,36], activity monitoring (e.g., fitness) [37], biometric authentication [38], and stress detection [39,40]. Such applications are integrated conveniently, affordably, and safely into daily life. In addition to physiological monitoring, psychological conditions can also be evaluated, for example in the car, to prevent accidents [35,36,41]. In addition, camera-based vital sign monitoring with mobile phones supports telehealth in rural regions, where clinics and specialized medical equipment are scarce [42,43]. 

The majority of work focus imaging PPG (iPPG, PPGi, PPGI), which is also named camera-based PPG, remote PPG (rPPG), distance PPG, noncontact PPG, or video PPG [15,19,23,44,45,46,47]. Classically, rPPG also stands for reflective PPG [15,48]. In the following, we therefore use iPPG for all image-based PPG measures. The HR and the HR variability (HRV) as well as the pulse rate (PR) and its variability (PRV) derive from ECG-based as well as PPG-based measures, respectively. Several authors such as Bánhalmi et al., confirmed the good correlations between HRV and PRV [49]. Therefore, we use the term HR and HRV to address both of the techniques. 

iPPG measures subtle skin color variations or cyclic movements of the body to estimate HR, RR, BP, BST, or SpO_2_. However, the field of camera-based vital sign measurement comprises not only red–green–blue (RGB, wavelength: 350–740 nm) receptors [7,50] but also infrared cameras, usually in near-infrared (NIR, 740–1000 nm) [51,52,53], and far-infrared (FIR, >1 μm) [15,25,29,34]. They measure the absorbance, reflectance, or transmission of light [43]. For instance, RGB cameras capture the reemitted light after passing through various skin tissues, while hemoglobin and melanin change the optical skin properties. The captured light may reflect the cardiac-related parameters that can be derived from variations of the intensity pixels in images. In addition, the respiratory-based component is based on the detection of periodic chest movements caused by the lungs volume changes while breathing. This method is reliable, safe, and affordable as it does not require special equipment, and it is used for long-term monitoring [21,54]. However, it is inapplicable in dark or low-light conditions [55]. 

FIR is also known as thermal imaging or infrared thermography [15,56,57]. It detects the temperature fluctuations around the nostril area as the temperature during inhalation and exhalation is similar to that of the external environment and the human body, respectively [15,32]. The cyclical ejection of blood from the heart to the head, through the carotid arteries and thoracic aorta, which results in periodic vertical motions of the head, may be used to monitor HR through utilization of FIR [57]. In addition, body skin temperature (BST) is a noncontact and noninvasive method of measuring the temperature continuously by the detection of infrared radiation emitted from the body. It can be monitored using FIR. FIR measures this infrared radiation, which results in estimating BST [58]. Therefore, it is suitable during the night in applications such as sleep monitoring, neonates, or ICU monitoring [25,59]. In addition, it has an advantage under varying illumination conditions and can help with privacy concerns [32]. However, most of the FIR cameras are high-end products and expensive [27], whereas consumer-accessible FIR cameras are challenged by low pixel resolution and sampling rates [60]. The NIR camera overcomes such issues.

NIR cameras work similar to RGB cameras, but they can also operate at night and in dark conditions if infrared (IR) lamps illuminate the scene [53,61]. Operation without visible illumination is advantageous for applications such as driver monitoring [36] and sleep studies [52,53,62]. The drawback of NIR cameras is the lower absorption by hemoglobin and a lower signal-to-noise ratio (SNR) [36,41]. However, camera-based vital sign monitoring faces many challenges associated with the hardware, motion artifacts, variability of natural illumination, skin variance, and the camera-subject distance. These factors negatively affect the accuracy of vital sign measurements [59].

In recent years, several iPPG review articles focus on HR measurement [50,63,64,65]. Zaunseder et al., focused on HR and HRV, as well as other derivable parameters including pulse transit time (PTT) and pulse wave velocity to remotely examine the peripheral vascular system [66]. Addison et al., explored RR measurements using depth cameras [67]. Maurya et al., analyzed noncontact RR monitoring of newborn using RGB-, IR-, and radar-based modalities [28]. Rehouma et al., comprehensively surveyed RR measurements using noncontact sensors, including cameras [68]. Khanam et al., reviewed motion- and color-based methods and focus on motion artefacts, illumination and distance variations, different sensors and acquisition settings, and demographic parameters [69]. Recently, Steinman et al., highlighted BP measurements utilizing smartphone technologies and video cameras [70]. Haford et al., presented an interesting review of demographic parameters, camera parameters, study design, and obtained accuracy on the reported vital signs such as HR, RR, SpO_2_, and BP [71]. Antink et al., provided an overview of camera-based measurements of vital signs using RGB and IR cameras [13]. However, existing reviews disregard how the vital signs are extracted from the camera frames. Most review articles focused on either HR or RR measurement using RGB cameras. Only a few considered BP, BST, and SpO_2_. 

To fill this gap, we present a complete review of recent advances in camera-based techniques for all five vital signs. We further aim to identify challenges of existing works, get insight in how to address these challenges, and identify future research prospects. In particular, we want to answer the following research questions:Which vital sign is monitored using what type of camera?What is the performance and which factors affect it?Which health issues are addressed by camera-based techniques?

## 2. Methods

We performed the literature review according to the preferred reporting items for systematic reviews and meta-analyses (PRISMA) workflow [72].

### 2.1. Selecting Databases

In previous work, we have identified PubMed, Scopus, and the Association for Computing Machinery (ACM) digital libraries as most relevant [73]. We searched these databases for articles published in the English language between January 2018 and April 2021. 

### 2.2. Composing the Search Query

Our main purpose is to retrieve the relevant literature on measuring the vital signs continuously, without contact, and remotely using different camera modalities. We developed the search string using relevant terms (Section A.1, Section A.2 and Section A.3 for PubMed, Scopus, and ACM, respectively). Relevant terms address the 

*Sensor technology*: camera, video, image, RGB, infrared, thermal, thermography;*Sensor setting*: remote, noncontact, contactless, contact-free, touchless, vision-based;*Biosignal*: biosignal, biomedical signal, physiological signal, cardiorespiratory signal, heart rate, heartbeat, pulse rate, respiratory rate, breathing rate, blood pressure, body temperature, oxygen saturation; and the*Modality*: photoplethysmogra* (to include photoplethysmogram, photoplethysmograph, and photoplethysmography).

### 2.3. Inclusion and Exclusion Criteria

We include and exclude articles according to:
Inclusion criteria
a.English languageb.Human researchc.Journal, Conference
Exclusion criteria
a.Review, systematic review, survey

From the retrieved articles, we first eliminated duplicates. The selection process consists of three stages: title, abstract, and full-text screening. In title screening stage, we removed work focusing on aspects not associated with vital sign measurement (e.g., face detection without vital signs monitoring and facial expression analysis) and work using other sensors (e.g., spectroscopy, radars). Subsequently, we performed the same selection process to the abstracts and to the full text. After literature analysis, we identified population parameters (e.g., age, number of subjects, subjects’ skin tone), camera parameters (e.g., camera, frame rate, resolution), applied algorithms, performance metrics, and predicted vital signs.

## 3. Results

Figure 1 depicts the PRISMA workflow and the results of the literature screening. A total of 1031 articles were found; from those, 124 duplicates were removed. The titles and abstract of the remaining 905 articles were screened and 781 were considered irrelevant. Subsequently, the full texts were examined with 19 being found to be irrelevant for this review and one full-text was inaccessible. To that end, we include 104 articles, all of which met the following eligibility criteria: Optical image (RGB, NIR, or FIR)Extraction of vital sign (HR, RR, BP, BST, or SpO_2_)Comparison to ground truth

Each year delivered 20 to 35 records. Researchers mainly focused on HR and RR, followed by BP, BST, and SpO_2_ (Table 1). The following subsections presents the results of the literature review. They are organized in the following order: data acquisition, image processing, signal processing, and performance metrics (Figure 2). At first, video or image data from smartphones, webcams, or digital cameras is acquired with synchronously obtained ground truth (Section 3.1). Existing datasets are also used (Section 3.2). As shown in Section 3.3, image processing techniques are then employed to detect the region of interest (ROI) and the raw signal is extracted from the ROI. In Section 3.4, we discuss signal processing techniques, mainly deep learning (DL) approaches, to remove motion and illumination artifacts. After optional data fusion (Section 3.5), the vital sign is extracted from the signal (Section 3.6) and compared to the ground truth using performance metrics (Section 3.7).

### 3.1. Data Acquisition

The data used in the analyzed articles was acquired in indoor (e.g., laboratory, hospital, home), or outdoor environments (e.g., in-car). Twenty articles used videos from existing datasets [52,59,74,75,76,77,78,79,80,81,82,83,84,85,86,87]. In the following, we describe the main findings regarding study design, hardware setup, and ground truth (Figure 3) that is presented in Appendix A.

#### 3.1.1. Study Design

The study design covers aspects from study population, namely number of subjects, age, gender, skin color, and the various types of settings. Researchers mainly recruited volunteers. Frequent populations were healthy [12,37,39,46,51,60,88,89,90,91,92,93,94,95,96,97], ICU patients [29,98,99,100], neonates or infants [24,25,60,98,99,101,102]. The majority of the articles obtained the data on their own (Figure 4). Number of subjects was mostly between 10 and 30. 

Regarding the age, study subjects were between 18 and 85 years old [53,60,90,103,104,105,106,107], while newborns aged from a few days up to 40 weeks [24,25,26,98,99,102,108,109] with different gender. Thirty-two articles reported their skin color. The Fitzpatrick scale (I to VI) was most commonly used to determine the skin color of subjects (8 articles) [43,60,92,101,110,111]. Visually resemble colors like white, pale, yellow, brown, or black skin reported in six papers [20,87,95,112,113]. In 18 papers, the subjects’ country of origin or geographical locations were indirectly used as skin color indicators [24,39,51,59,75,85,90,98,99,104,109,114]. 

#### 3.1.2. Hardware Setup

In the reviewed articles, RGB, NIR, FIR, or a combination of RGB with NIR or FIR camera were used (Figure 5). RGB cameras such as smartphone cameras [43,49,104,115], webcams [20,95,116,117], or digital cameras [56,78,102,108] were utilized often, due to their low costs and easy availability. Further, most researchers use single RGB camera, followed by combination of cameras. 

Several parameters affect the image quality of cameras, such as spectrum range, sensor hardware, frame rate, frame resolution, or camera–subject distance. Frame rate ranges from 1 frame per second (fps) to 500 fps. For fast moving subjects, frame rates of 100 fps to 500 fps were employed [45,49,118,119], however 10–30 fps was used often. Rapczynski et al., reported that higher frame rates do not contribute more information but increase demands for storage space, transmission, and computation instead [59,80]. Applied resolutions vary greatly, ranging from 2080 px, down to 47 px. Although different ranges of resolutions are observed in articles, a resolution of 320 pixels (px) × 240 px is sufficient to recover a signal with an adequate SNR [120]. However, the 480 px to 720 px range widely used. 

In addition, environmental (e.g., illumination, temperature) and behavioral (e.g., body movement) parameters impact the image quality [117]. For charge coupled devices (CCD) cameras, illumination is one of the key parameters that is provided by sunlight through windows [59,95], ambient illumination from ceiling fluorescent lights [46,87,101,102,107,110,121], phone flash lights [12,43,49], uniform white light [37,122,123] or combination of ambient light and sunlight through windows [81,86,109].

Figure 6 shows the conceptual semantic relationship between camera images and vital signs. The majority of articles was estimated the HR by utilizing RGB images. Similar to HR, RR measurements were calculated using both RGB, NIR and FIR images. Measurements of BP and SpO_2_ were extracted from RGB images. FIR images were used to calculate BST only. None of the research used RGB images to estimate BST based on the author’s knowledge.

#### 3.1.3. Ground Truth

In addition to the camera-based modalities, all research employed typically assessable or commonly available measurement devices to record the vital sign’s ground truth (Figure 7). Fifty-five articles used manual palpation (e.g., radial artery), ECG, or pulse oximeters to record the HR [39,49,86,95,102,107,110,112]. Sixteen articles utilized manually counting (e.g., visual inspection), respiratory belt, pressure sensors, or polysomnography to calculate the RR [32,51,60,89,94,105,106,124,125]. Six articles measured BP by utilizing manual sphygmomanometers or a blood pressure cuff [37,113,119,123,126]. Two and one articles used clinical thermometers and pulse oximeter that delivers the reference for BT and SpO_2_, respectively [61,127]. Nine articles utilized patient monitor, or wrist-based wearable devices to track vital signs (e.g., Philips patient monitor) [20,25,93,99,102,108,128,129,130] and fifteen articles delivered multiple vital signs measurements [43,51,53,57,94,106,122,131,132].

### 3.2. Existing Datasets

Publicly available datasets (Table 2) have been recorded while the subjects are watching video stimuli [133], watching music videos [134], playing a stressful game [135], moving their face (e.g., speaking, slow and fast rotation) [136], using various illuminations [137,138] and during driving [41]. Different cameras captured the ROI in the VIPL_HR [137], TokyoTech Remote PPG [118] and MERL-Rice near infrared pulse dataset (MR-NIRP) [41]. All experiments were conducted in an indoor setting except MR-NIRP. The frame rate varies between 20 fps and 60 fps. However, the majority of the databases captures video with RGB cameras at a frame rate of 30 fps. Along with camera-based modalities, typical clinical parameters such as HR, RR, BT, and SpO_2_ were obtained. All databases are available upon the approval of the end user license agreement except publicly accessible MR-NIRP database.

### 3.3. Image Processing

Once the video data is available, it is fed into a typical pipeline for image-based vital sign measurements (Figure 2). Usually, image processing contains ROI detection, tracking, enhancement, color decomposition, and raw signal extraction.

#### 3.3.1. ROI Detection

Depending on the parameter of interest, the ROIs differ. For instance, the ROI for HR measurements is the location of the facial artery (e.g., cheeks) [46,59,79,151], while RR often is extracted from the nasal or torso region [29,31,91,128,152]. The orbital region and the full frame are the preferred ROIs to calculate the BST [61,100,127,132]. Forehead, forearm, palm, and cheeks are included to measure BP [37,113,119,126].

Figure 8 depicts the workflow of the HR vital signs measurement from different camera images. These articles considered the face [83,85,87], forehead [20,43,46,102], cheeks [75,79,132,153], nose [40,94,109], chin [40], and palm [37]. Furthermore, four articles partitioned the entire frame or face into sub-ROIs and select the most reliable sub-ROI [27,77,91,119]. Having more capillaries and being unaffected by facial expression, the forehead and the cheeks are most commonly extracted as ROI [38,95,110,130,151,154]. 

Different algorithms identify the ROI either manually [29,57,101,102,105,111,125,128,152] or automatically, where the Viola–Jones algorithm is used most frequently (Table 3). In addition to the listed articles, four articles employ the discriminative response map fitting algorithm to locate the face landmark points [77,87,107,153]. Addison et al., detected a seed point on the forehead and the skin around it and performed a flood fill for each frame that recursively aggregates all adjacent skin pixels within the preset tolerances [110]. Recently, neural-network-based ROI detection has gained increased attention and yielded higher accuracy compared to conventional approaches [78,155]. 

#### 3.3.2. ROI Tracking

ROI extraction can be done by detecting the ROI in all frames or by identifying ROI in the first frame and tracking over all frames [20]. Sixteen research teams used the Kanade–Lucas–Tomasi algorithm [29,31,36,59,60,76,85,89,107,122]. It is also effectively tracks the ROI in NIR images [156]. Alternate tracking using kernel correlation filters show false detections in out of sight targets [75,109,157,158]. Liu et al., employed the compressive tracking algorithm to detect multiple subjects [114]. Hochhausen et al., and Pavlidis et al., utilized the particle filter-based object tracking algorithm in thermal images, but this method requires a multicore computer system [132,152,159]. 

**Table 3 sensors-22-04097-t003:** Automatic detection of ROI by utilizing various classical and DL methods.

Algorithms	Description	Advantage	Disadvantage
Viola–Jones [18,75,81,83,87,93,106,115,129,153]	It utilizes Haar-like features and Adaboost algorithm to construct a cascade classifier.	It works well on full, frontal, and well-lit facts.	It suffers from faces in a crowd, face rotation, inclined or angled faces, expression variations, and low image resolution.
Histogram of oriented Gradients [86,151,160]	It constructs the feature by calculating the gradient direction histogram on the local area of the image.	Fast running speed and identifies 68 facial landmark points.	It may be influenced by light intensity and detection and inaccurate location of feature points on profile.
Multitask cascaded convolutional neural network [154]	It is a convolutional-neural-network (CNN)-based framework, which consists of three stages for joint face detection and alignment.	Accurate face detection, less affected by light intensity and direction.	It may provide sophisticated models and calculation, which may result in a slow running speed; only five feature points can be tracked.
Single shot multibox detector [78]	It is a fast convolutional neural network to detect faces using a single neural network.	Fast processing speed and multiscale feature map is adopted.	The robustness of the network to small object detection may not too high.
You look only once [100,161]	It is one stage detector based on object detection.	Fastest object detection algorithm. It utilizes full image as context information which is possible to achieve real-time requirements.	It requires a graphics-processing-based computational machine. It may be relatively sensitive to the scale of the object.
Template matching [51,61,106]	It matches the image by providing a base template which to compare.	Relatively easier to implement and use.	Not suitable for complex templates, no face in the frame, or occlusion of face.

#### 3.3.3. Image Enhancement

Enhancement of low-resolution images improves the resulting vital signs. For instance, subtle color changes in the ROI are difficult to measure reliably. The temporal magnification approach relies basically on two main types: Lagrange and Euler [98]. The Lagrange technique tracks the mobility of each pixel, whereas the Eulerian approach splits the image into grid elements and tracks the movement of pixels [98,162]. So far, eight research teams utilized Eulerian video magnification to magnify the subtle movements [24,125]. Moya-Albor et al., applied the Hermite transform-based Eulerian video magnification technique with improved accuracy [97]. Furthermore, Kwasniewska et al., and Yu et al., introduced the super resolution-based deep neural network to improve the performance of vital sign estimation of hallucinated thermal image [125] and the spatiotemporal video enhancement network, respectively [163].

#### 3.3.4. Color Channel Decomposition

Depending on the color space, channel decomposition separates colors or intensity and illumination. Most commonly, researchers use RGB; hue saturation value (HSV); and luminance, chroma blue, chroma red (YCbCr). Sixty-five articles explored RGB channels [37,95,104,110]. Green channel contains more information related to cardiac pulse [38,104,130]. Pai et al., mentioned the intensity of red and blue channels to eliminate illumination variations [164]. Only two articles regarded the red channel [115,151].

The HSV color space has been explored by three research teams [43,114,130]. HSV does not require color normalization and is robust to illumination variations [154]. Sanyal & Nundy explored the hue channel to measure the fluctuations associated with skin color changes and used for iPPG signal extraction [43,114]. Bánhalmi et al., examined luma component from YCbCr [49]. Ryu et al., combined chroma blue and chroma red channels providing a high SNR [142]. YCbCr and HSV-based skin-representing pixels have been simply compared with a threshold [81,98,116,154]. In addition, Li et al., Rapczynsk et al., and Barbieri et al., employed the Conaire’s method [85], skin probability-based approach [80], and a decision tree classifier [92] to segment the skin pixels, respectively.

#### 3.3.5. Raw Signal Extraction

Averaging the ROI in each frame extract the one-dimensional iPPG signal that results in reduction of quantization noise [54,142,164]. The majority of research (94 articles) articles spatially averaged to obtain one-dimensional signal [29,46,59,60,79,107,109,115,152]. On the other hand, several studies such as Zhu et al., utilized feature points to track the cyclic motions of the ROI [165]. 

### 3.4. Signal Processing

Signal processing usually conducts the preprocessing of the raw signal and vital sign extraction.

#### 3.4.1. Preprocessing

Obtained from spatial averaging of ROI pixels, the vital signs’ raw signals comprise noise from subject movement, facial expression, illumination variation, and dynamic environments. Hence, before extracting vital signs, the raw signals need appropriate pre-processing (Table 4).

The critical parameter of a band-pass is its bandwidth. According to the Association for the Advancement of Medical Instrumentation (AAMI), the bandwidth should be in between 30 bpm to 200 bpm for HR studies in human adults [168]. However, HR varies depending on a variety of factors including age, exercise, medications, and medical conditions [6]. Therefore, different filter bandwidth settings have been applied such as, 0.3–6 Hz [92], 0.6–2.8 Hz [93], 0.6–3 Hz [114], 0.75–3 Hz [59], 0.5–4 Hz [110], 0.7–4 Hz [79,130,160], 0.65–4 Hz [116], 0.83–3.17 Hz [122], and 0.8–2.2 Hz [43]. Since infants under the age of one year have a higher HR than adults, ranging from 110 to 160 bpm [98,169], research on infants reported frequencies of 1.83–2.67 Hz [108], 1.3–5 Hz [109], and 1.67–3.33 Hz [101].

The RR of adult lies between 12 cpm and 20 cpm [170,171]. The RR varies with factors such as age, exercise, and respiratory problems. The researchers have considered the frequency ranges of 0.05–1.5 Hz [32], 0.18–0.5 Hz [43], 0.05–2 Hz [105], and 0.1–5 Hz [89]. Similar to HR, infants have a higher RR than adults, ranging from 40–60 cpm [27]. The bandwidths of RR were in the range of 0.1–3 Hz [27], 0.5–1 Hz [108], and 0.2–2 Hz [102]. Band-pass filters in BP studies have the bandwidths of 0.7–2 Hz [37] and 0–6 Hz [172]. With respect to SpO_2_, researchers have used a bandwidth of 1.67–3.33 Hz [101] and 0.5–5 Hz [12].

Although motion artifacts are a key problem while sleeping as well as in infant studies, only a few articles address this problem. Benedetto et al., removed motion artifacts manually [39]. Eight studies identified body motion threshold-based [32,37,90,95,115,153,173]. Lorato et al., applied a gross motion detector to remove torso and chest motions [26]. In neonates, Rossol et al., used a motion history image algorithm [24]. 

#### 3.4.2. Vital Sign Extraction

According to the various vital signs, researchers conduct several techniques to extract the vital signs from the raw signals. 

##### Heart Rate

The video-extracted signal is composed of a component representing the blood volume pulse as well as a noise component. The dimensionality reduction is commonly categorized as blind source separation (BSS), such as independent component analysis (ICA) and principal component analysis (PCA), model-based methods, and DL, that is employed to extract pulse related information [168] (Table 5). BSS extracts the source signals without previous knowledge. Further, ICA may require more computation to separate the components [174]. To overcome this problem, the variants of ICA, namely FastICA [154], viterbi decoding-based ICA [83], project_ICA [174], and joint BSS along with ensemble empirical mode decomposition [157] are used by Zheng et al., Raseena & Ghosh, Qi et al., and Lee et al., respectively. On the other hand, Lewandowska et al., employed a PCA in 2012 [175]. Barbieri et al., reported result on the zero-phase component analysis, a variant of PCA, to find the peaks with more precision [92,162]. However, according to Huang et al., BSS-based methods leave the skin color information unutilized [168]. Chrominance (CHROM)-based [162] and plane-orthogonal-to-skin (POS) are traditional model-based methods. It provides more stable under complicated and nonstationary circumstances [168]. 

Recently, DL techniques gained a great amount of interest which are categorized as non-end-to-end and end-to-end. Non-end-to-end techniques apply DL either in image or in signal processing [181]. Hsu et al., employed CNN to extract HR by applying two-dimensional time–frequency representation [78,145]. Qiu et al., utilized spatial decomposition and temporal filtering to extract feature images and cascaded them with a CNN [75]. Moya-Albor et al., trained CNN by feeding magnifications from a Hermite transform [97]. On the other hand, end-to-end learn features on their own to generate the vital sign that is employed by a few works and described as follows: Yu et al., used spatiotemporal video enhancement networks and the rPPGNet to implement end-to-end HR extraction without utilizing classical algorithms [163]. Huang et al., demonstrated two dimensional convolutional along with long short-term memory to extract HR [182]. Further, Perepelkina et al., utilized HeartTrack that is constructed by a CNN [121]. 

In addition to HR, other cardiac-based metrics, such as HRV and interbeat interval (IBI), are used to assess physical and mental health in daily activities [40,164]. The sympathetic nervous system, which regulates blood vessel diameter, is inversely correlated to HRV [172]. HRV calculates the variations between the heart beats, whereas IBI measures the time interval between the two consecutive heart beats [56,104]. According to the joint Task Force of the European Society of Cardiology and the North American Society of Pacing and Electrophysiology, the HRV may be evaluated either in the time or in the frequency domain [104,183]. Typical parameters in the time domain are standard deviation of interbeat intervals, root mean square of successive differences in interbeat intervals, proportion of successive normal-to-normal that differ by more than 50 ms, and mean of PP interval [40,93,104,118]. Frequency domain features are very low frequency components, low frequency components, high frequency components, and the ratio of low frequency to high frequency [92,104]. These features are analyzed statistically. Typically, researchers perform filtering or decomposition of the signal before extracting these features in the time or frequency domains [85,92,95,117,118]. 

##### Respiratory Rate

RGB cameras detect the RR from breathing motion, whereas IR cameras identify both temperature fluctuations caused by breaths and motion [57,131]. FIR camera resolution varies strongly (Table 6). In RGB images, dimensionality reduction methods, namely PCA, zero-phase component analysis and POS, were applied by Zhu et al. [53], Iozza et al. [89], and Chen et al. [56], respectively. Recently, Schrumpf et al., utilized empirical mode decomposition to extract the respiratory related component from intensity-based signal. It decomposes the signal into numerous intrinsic mode components and identifies the relevant component using power spectral density [91]. 

RR was monitored using thermal camera in fifteen articles. In this, eleven articles showed that inhaling and exhaling at regular intervals causes a proportional decrease and increase of the temperature around the nasal cavities, respectively, which can be converted into RR [29,31,60,94,125,128,152]. Three articles investigated periodic motions of chest in the full frame [25,26,27]. 

##### Blood Pressure

A common approach of calculating BP is to extract waveform-related features such as temporal, areas, amplitude, and derivative features by utilizing systolic and diastolic peaks [46,88]. Sugita et al., extracted the fluctuations of BP waveform from palm ROI [37,186]. However, such indirect methods cannot observe the BP absolutely. The PTT is a valuable metric for tracking variations in BP throughout regular activities [186]. It is possible to obtain by calculating the time taken for an arterial pulse to travel along an arterial segment by monitoring at two different sites on a blood vessel [111,119]. It can be seen from the difference between vessels near and far from the heart. Shirbani et al., measured the PTT between the forehead and left-hand ROI of the iPPG signal [111]. The time difference from two ROIs codes the blood pressure [186]. On the other hand, studies employed the signal processing techniques to extract hemoglobin related signal features that are used to estimate BP. Zhou et al., applied ICA and peak detection to identify the peak and valley of the wave in order to estimate the systolic BP and diastolic BP respectively [113].

Stimulus (e.g., food, exercise activities) has an impact on the sympathetic nervous system and elevates the BP [122,126]. Zhou et al., conducted the test during rest condition [113]. Sugita et al., and Takahashi et al., conducted the experiment during exercise activities (e.g., squat exercise) [37,119]. Vasoconstriction is the narrowing (constriction) of blood vessels by small muscles in their walls, which was stimulated by Oiwa et al., via cold temperature [126]. These experiments conducted in laboratory settings, used facial or hand/palm ROIs. We have not identified any study in a real-world setting. 

##### Body Skin Temperature

Six articles acquired thermal videos to measure BST by obtaining the highest temperature of forehead [61,132], or the entire frame [94,100,127,131]. Dagdanpurev et al., employed the *K*-nearest neighbor algorithm to differ healthy and infected patients [127]. We did not find any method to derive BST from RGB. 

##### Oxygen Saturation 

Forehead and hand backside are common ROIs to measure SpO_2_ from RGB images. Classically, the SpO_2_ is calculated from the ratio of absorbance at red and blue channels (ratio-to-ratio). It is based on the effective blood volume and the hemoglobin concentration [12]. However effective blood volume can be influenced by peripheral vasoconstriction, local temperature, or cardiac index. To overcome this problem, Sun et al., explored multiple linear regression models [12]. Wieler et al., extracted the oxygen desaturation from the facial region of the forehead in neonates [101]. From this region, red and blue channels obtained to calculate the peak-to-trough ratio of two wavelengths. It corresponds to the maximum and minimum absorption of oxygen in blood and is an indirect measure of peripheral SpO_2_ [101]. 

### 3.5. Data Fusion

To overcome limitations by a singular data source, some articles fuse data from different sources. We differ image-based and signal-based fusion that uses several video sources and combines the data already on a frame-based level and after the raw signals have been extracted separately from the different videos, respectively. For example, Lorato et al., merged multiple thermal camera images into a single image before computing RR and automatically detect the core pixel that corresponds to the respiration [25]. Villarroel et al., combined patient detection and skin segmentation from an optical flow network [99]. Kurihara et al., combined RGB and NIR face patches to estimate the HR, which is helpful in dark environments and eliminate background illumination [187]. McDuff et al., delivered an intriguing investigation that involved the use of nine distinct cameras with diverse angles for HR estimation [120].

On the other hand, Kado et al., used histogram fusion to select suitable face patch pairs in HR measurements, based on thresholding [55]. Pereira et al., explored three different sensor fusion strategies: median frequency of all valid ROIs, highest signal quality index, and Bayesian fusion [27]. Scebba et al., introduced a signal quality-based fusion approach to calculate the RR by determining the signal quality of various respiratory signals such as thermal airflow and respiratory induced motion. They also used S^2^ fusion, a combination of signal quality-based fusion with apnea detection to obtain the RR [60]. Fujita et al., implemented a CNN to select the spectrum from a set of multiple obtained ROI from a single frame [159]. 

In summary, fusion-based techniques can significantly improve the robustness of vital sign monitoring, despite the motion and illumination problems in real-time situations [15]. For instance, sensor fusion techniques that employ multiple cameras positioned at different angles can reliably track the ROI of the subject and compensate motion artifacts (e.g., head turn). A combination of RGB and NIR cameras for day and night environments is helpful in low light applications [187]. Furthermore, signal fusion might allow single camera approaches to assess multiple ROIs simultaneously. 

### 3.6. Vital Sign Estimation

The conversion of the signal into a vital sign can be done in the time (34 studies) or in the frequency domain (59 studies). In the time domain, peak detection has diffusely been applied to obtain the IBI from which vital signs are computed. Peak detection may reduce the error by averaging all the IBI over the video duration [32,40,77,105,115,116,118,151]. In the frequency domain, a global or windowed (short-time) Fourier transform may identify the peaks with maximum power, which then are simply turned into a rate of a vital sign [59,60,78,82,94,117,122,130].

### 3.7. Performance Assessment

#### 3.7.1. Performance Metrics

The mean error and its standard deviation, MAE, mean relative error, mean of error rate percentage, limits of agreement, RMSE, CC, and correlation of determination assess the performance of the various algorithms. 

##### Heart Rate

Figure 9 depicts the various conditions investigated in HR measurements. Studies conducted in static conditions resulted in low RMSE [49,95,107,111,114,130]. Dynamic conditions such as subject movement, speaking, facial expressions, and exercise activities provided less reliable results [82,87,107,130,156] (Figure 9a). HR measurements perform at various distances 0.15–3 m throughout the studies (Figure 9b). Less than 1 m yielded satisfactory results but the RMSE increased with larger distances between the subject and the camera [59,87]. Outliers are due to movements [94]. Five research teams particularly investigated on distance variance [59,83,87,156,188]. For instance, Zhang et al., studied 40–80 cm and reported that 60 cm provides the optimal performance [156].

Similarly, varying the intensity of the illumination causes a different result. At static conditions, light intensities of 150 lux and 300 lux yielded RMSEs of 5.56 bpm and 5.58 bpm, respectively. Dynamic conditions, such as rotation, mixed action, four feet walk, reached RMSEs of 8.09 bpm, 9.96 bpm, 8.60 bpm at 150 lux and 7.36 bpm, 9.88 bpm, and 7.72 bpm at 300 lux respectively [130]. Hassan et al., varied intensity levels such as 340–380 lux, 430–470 lux, and 510–550 lux to check the impact on HR estimation, which resulted in 20.27 bpm, 16.86 bpm, and 13.98 bpm, respectively [87]. Hence, it shows that increasing the illumination reduces the error in RGB. Low-light (1 lux) and bright illumination (184 lux) conditions by utilizing NIR light resulted in RMSE of 1.06 bpm and 0.86 bpm [51]. Zhang et al., varied the intensity levels for the driving condition between 0 lux and 80 lux and demonstrated with webcam and NIR video. The performance deteriorated if the intensity level is less than 20 lux in webcam video, whereas NIR video provides better result even at 2 lux. Zheng et al., also reported the optimal illumination at 40 lux for normal driving situations [156]. However, drastic illumination changes might occur during driving due to the streetlamps and headlights [36]. In summary, HR measurement using RGB cameras suffer from darkness, whereas NIR cameras yield better results. 

Variation of skin tone (e.g., fair, brown, black) also influences the performance. Under static condition, the RMSE for fair skin was 0.07 bpm [95]. On the contrary, the RMSE for fair skin with limited subject movement condition is 17.75 bpm [87]. Song et al., investigate Asian subjects at 1 m distance and 2704 px × 1520 px, reaching 1.50 bpm in static condition [59]. The Chinese subject group attains an RMSE of 2.77 bpm with minor mobility [114]. The RMSE of 17.10 bpm reaches with brown skin and minimal physical activity [87]. For dark skin, participants in stationary state yielded an RMSE of 16.49 bpm [87]. Addison et al., reports the RMSE of 1.74 bpm, 2.32 bpm, 1.88 bpm, 2.04 bpm, 2.89 bpm, and 2.43 bpm on the Fitzpatrick scales I to VI, respectively [110]. From these studies, we can conclude that increased melanin concentration lowers the performance of HR estimation.

##### Respiratory Rate

Researchers measured the RR in different conditions: normal and simulated breathing with controlled frequency [27,60,89,91], different clothing [105], and different camera-subject distances [29,52]. Studies conducted in stay-still situations yielded better results (Table 6) [27,29,105,124,128,152]. In RGB images, Chen et al. [56] and Sanyal & Nundy [43] performed a steady state with minimal movement and yielded an RMSE of 2.16 cpm, and 3.88 cpm, respectively. In NIR images of sleeping subjects, Zhu et al. achieved an RMSE of 2.10 cpm [53]. Chen et al., utilized NIR in a still state with varying illumination and reached an RMSE of 0.70 cpm [51]. In a regular setting, Pereira et al. [27], Negishi et al., [106], Kwasniewska et al. [125], and Cosar et al. [94] used thermal images and obtained an RMSE of 0.31 cpm, 1.44 cpm, 3.40 cpm, and 3.81 cpm, respectively. In contrast, movements lower the performance down to 2.52 cpm and 6.20 cpm as shown by Negishi et al. [106] and Cosar et al. [94], respectively. From all these findings, the median RMSE is 2.1 cpm and 3.9 cpm for distances d ≤ 1m and d > 1 m, respectively (Figure 10). Hence, RMSE grows with increasing distance in the same way as HR does [29].

Real time applications for neonatal care involve both HR and RR measurements [24,25,26,27]. HR measurements of inactive infants achieved a RMSE of 1.1 bpm [109]. However, infant studies provided larger limits of agreement due to motion and unconsciously swinging hands in HR and RR measurements [24,98,102,108]. 

##### Blood Pressure

Zhou et al., reported RMSE of systolic and diastolic BP without movement in a distance of 60 cm as 6.91 mmHg and 4.88 mmHg, respectively [113]. Additionally, they varied distances (30–90 cm) and illumination (75 to 500 lux). Substantial inaccuracies arise when the distance is large and the ambient light is bright or faint. Zhou et al., reported best results with a light intensity of 150–200 lux and a camera-subject distance of 50–60 cm [113]. Sugita et al., achieved RMSE of 25.7 mm Hg during cycling exercise [37]. Studies with food stimuli (e.g., chocolates) [122] and cold stimuli [88,126] further affected BP. 

##### Body Skin Temperature

Dagdanpurev et al., reported significant results between healthy subjects and infected patients and a correlation of 0.82 during steady state [127]. Cosar et al., obtained an RMSE of 0.86 °C and 0.88 °C in steady state and moving conditions, respectively [94]. 

##### Oxygen Saturation

Sun et al., explored the peripheral SpO_2_ in healthy adults using smartphone cameras and a ROI from the back of hand [12]. They reported estimated errors less than 1%. Wieler et al., investigated the oxygen desaturation on healthy newborns with an RGB-based approach and reported 75% sensitivity and 20% positive predictive value [101]. 

#### 3.7.2. Factors Affecting the Performance

Facial hair and skin tone variation have a strong effect on vital sign estimation (Figure 11) [128]. In this, darker skin tones have a higher melanin concentration [87,111], which absorbs a large amount of incoming light, causes the signal to attenuate, and weaken the measured iPPG signals [104,164]. In addition, makeup weakens the pulse signal [96,120]. 

Disregarding the camera type, camera parameters such as resolution and frame rate always have a tradeoff [120]. If the image resolution is too low, quantization noise interferes the iPPG signal, that is difficult to reduce by spatial averaging in video frames. Although resolution is high, pixels within ROI may be insufficient at far recording distance [59]. Hence, increased camera-subject distance reduces the performance (Figure 9b). According to Nyquist theorem, a frame rate of 8 fps is required to acquire the data with a HR of 240 bpm and a RR of 90 cpm. However, studies most commonly used the frame rate of 30 fps for vital sign estimation. 

Meanwhile, as the amount of light received by the camera sensor decreases, the signal strength of the iPPG pulse also drops. As a result, the impact of video quality, particularly the seldom used ultrahigh definition must be studied at different camera-subject distances with diverse angles [59]. Motion artifacts are an important factor, too. Several techniques (e.g., ICA, CHROM) significantly reduce such artifacts but it may suffer from head orientation, fast moving, partially covered ROI, and emotions in real-time [41,82,87,168]. 

Last but not least, environmental parameters such as illumination and outside temperature influence the accuracy of iPPG. Sunlight or artificial light are the most prevalent sources. Sunlight provides almost white light and reliable measurements. In contrast, artificial light results in noisy measures [117,189]. High illuminations yield better performance [87,130]. However, fluctuations in illumination such as obtained in a driving vehicle are most difficult to handle and may also occur in indoor environments [45,117].

### 3.8. Applications 

iPPG is used in a variety of environments including clinical and nonclinical settings. Clinically, iPPG performs general health examinations on a regular or continuous basis [35,99,104]. Additionally, it is applied to patients in critical care units, the post-anesthesia care units, during surgery [29,31,128], and to newborns [25,27,98,99,101,102,108,109] as well as geriatric patients [23]. 

Nonclinical settings include in-home [33] and in-vehicle health monitoring [35], exercise tracking [190], sleep monitoring [53], and stress monitoring [40]. Home-based vital signs monitoring is extremely helpful in the current pandemic situation due to limited hospital space and facilities [33]. It is also useful for monitoring individuals of all ages, particularly geriatric patients [23] and newborns [102,108,109]. In the automotive area, physiological and psychological condition of the driver are monitored to avoid accidents [35,156,168]. Huang et al., additionally monitor vital signs of vehicle passengers [168]. 

Although fitness is beneficial to health, too excessive activities might have serious consequences. Hence, it is critical to monitor the physiological status during exercises [49], such as stationary cycling [37]. Sleep monitoring addresses the vital signs as well as its abnormalities, such as sleep apnea and heart disease [32,60,93]. In addition, stress and anxiety monitoring avoids panic attack and accidents during driving [40]. 

## 4. Discussion

We perceive that work done in this field within the short period of this review (January 2018–April 2021) is significant and it may open up new research directions. We also believe that noncontact vital sign monitoring has gained increased attention among researchers and physicians during the current pandemic situation. We limited to image-based techniques for the scope of this review and excluded non-image-based noncontact techniques. This section discusses on the review’s limitations and responses to the research questions. Further, we comprehensively explain the research gaps in these fields as well as future research directions. 

### 4.1. Limitations

In this paper, we have reviewed the state-of-the-art in vital sign monitoring using cameras. We searched PubMed, Scopus, and ACM databases for relevant articles that have been published during the last three years. We have screened 1031 articles and included 104. There might be other sources that deliver further relevant articles. Although we carefully selected our search terms (Appendix B), we meanwhile observed that our search cannot be reproduced anymore. The number of returns today differs from that of April 2021, as in particular Scopus has obviously updated its entries including more articles. This may be due to late record submissions to the databases or electronic first publication styles. We might have missed relevant articles due to the selection of our search strings. So, this review may not cover the complete literature.

### 4.2. Research Questions

We now answer our research questions:*Which vital sign is monitored using what type of camera?* Using RGB cameras, the iPPG signal is extracted from a ROI in the video frames to monitor HR, SpO_2_, and BP based on color intensity changes, whereas RR is monitored based on body motion. NIR cameras measure the HR and RR as similar to RGB. In addition, thermal cameras extract HR, and RR based on periodic motions of the torso area, breathe airflow, or vertical movement of the face, whereas BST is obtained from the highest temperature of the ROI (Figure 6).*What is the performance and which factor affects it?* Static conditions avoid physical movement in well-controlled environments. Here, error rates are less than 5 bpm and 3 cpm for HR and RR, respectively (Figure 9a). The performance suffers considerably from body motion and face expressions and is further lowered if the camera-subject distance is more than 1 m (Figure 9b and Figure 10). In addition, changes in illumination considerably impacts the performance. Low light illumination decreases accuracy. These effects are quantified using several metrics (e.g., MAE, RMSE, CC).*Which health issues are addressed by camera-based techniques?* The working horses’ applications are ICU patients, neonatal, and geriatric monitoring as well as daily monitoring at home and hospital, during driving, and while performing exercises. Further, since March 2020, the COVID-19 pandemic is ongoing globally. As the virus affects the lungs, noncontact SpO_2_ monitoring is a valuable alternative to contact-based methods, as it avoids transmission of the virus. HR, RR, and BST can also be monitored to keep the best track of subject’s health. It is also further useful for monitoring vital signs in a noncontact way during other pandemic situations.

### 4.3. Research Gaps and Future Research Directions

In the following, we discuss the future of camera-based health monitoring and if camera-based approaches could eventually replace contact-based modalities in the clinical environment. First, the number of vital signs is limited with the vast majority of studies focusing on HR, RR, and BP [28,50,66,191]. Other measures are arterial stiffness and distal pulse reflection [88]. So far, BST cannot be measured using RGB cameras. Additionally, there is difference between body core temperature (BCT), BT, and BST. BCT refers to the temperature of the internal tissues of the body (e.g., pulmonary artery). Due to the balance of heat generation, absorption and loss, BCT maintains at constant temperature [58]. However, it can only be measured invasively by utilizing catheters. BT is a contact-based technique that utilizes sensors at various locations of the body (e.g., oral, axillary, rectal), which makes it difficult to use for infants or children. Body mass index, age, and the metabolic rate impact BT [192]. BST is an indicator of skin blood flow that may reflects the temperature of the body [58]. Current studies focus on BST estimation to measure core temperature from facial regions utilizing FIR imaging. However, physical activities and environmental factors differ the measure from the medically more relevant BCT. Obtaining BCT from the facial region using FIR is still under research, and machine learning might help here. 

Although the research on iPPG is relatively recent, it has rapidly spread from the laboratory to everyday environments research. As a substitute of conventional contact-based techniques, iPPG will be able to assess almost all physiological measurements that contact PPG provides. However, it remains under research due to various reasons that is discussed below in detail. Until now, only stationary studies in well-controlled settings yield reliable vital signs. The performance degrades significantly if the illumination varies, the subject has darker skin tone or is in motion, or if camera–subject distance is too large. In this, RGB images with proper illumination provide better signal performance, whereas, NIR cameras have a lower SNR due to the lower absorption of hemoglobin [41]. However, a few studies recently concentrated on NIR camera-based vital sign monitoring owing to the advantages of functioning in dark conditions (e.g., sleep monitoring, driver monitoring) [34,41,93]. Even though, it is not explored widely. Future research on NIR cameras might yield a path for camera-based vital sign monitoring in all light conditions.

We further are concerned that most studies and existing databases use a small number of participants (n = 10–30) and homogeneous groups (e.g., same skin group or particular geographical location). In addition, there might be bias in age and gender due to changes in the skin properties. There is no work focusing exclusively on healthy elder subjects or children above age of two. Hence, further studies are required to effectively evaluate the influence of age and gender. Most of the studies disregard skin colors although the melanin concentration has a strong impact on the measurements. Developing robust algorithms using DL might help here. Furthermore, the majority of the studies was conducted in indoor environments and obtained reliable results. Outdoor environments, on the other hand, can result in large amounts of noise, making it difficult to monitor vital signs [36,168]. This limits the translation into clinical practice and real-time environments. Though the techniques need further improvement before being used in clinical medicine, home-based indoor applications can be explored already [193]. Future research is required to focus on resolving the issues by tuning these parameters, namely camera–subject distance, camera parameters, and illumination, in various indoor and outdoor environments to obtain the best possible results that might improve the robustness in practical applications (Figure 12). As of today, subject and camera parameters are not mentioned in most of the article. Researchers need to improve and follow existing guidelines such as STARE-HI [194]. 

On the other hand, a few studies have shown to yield stable results even under nonstationary conditions [75,76,115,121,157,190]. However, most of the studies suffer from motion artifacts due to the dynamic situations. Although a band-pass filter extracts an interested frequency region, utilization of motion artifact-based algorithms will help to reduce the error. But only a few studies employed them so far [24,26,37,49,89,190]. In addition, the conventional signal processing methods, such as parametric-based BSS method and model-based methods, reported in several articles to recover the pulse signal, may fail due to lack of previous knowledge and assumption of skin tone vectors, respectively. However, nonparametric-based fast kernel density-ICA called semi-BSS, presented by Song et al., is reported to have promising results with varied resolution and camera–subject distance [59]. Although DL-based approaches have significant potential, they are still in the early stages of research for iPPG. However, end-to-end DL networks as demonstrated by Perepelkina et al., and Yu et al., could be a potential approach which have been shown to outperform classical signal processing techniques [121,163]. In the future, semi-BSS-based techniques and DL networks could be employed to overcome the classical signal processing methods. 

The majority of the articles relied on a single camera-based physiological monitoring, either RGB or IR. However, single cameras cannot persistently acquire the frames due to head or body posture variations or presence of obstacles. This might lead to missing observations or loss of data. Thereby, multi-camera-based researches could be able to overcome this challenge with the help of fusion techniques [120]. Studies such as Liu et al., detect HR from multiple people by employing a single camera frame and yield promising results [114]. Hence, combining these two methods, multi--camera and multi-person-based vital sign monitoring could be extended with the help of surveillance camera in hospital, in-home, and workspace environments. Our group recently explored these measurements for continuous health monitoring in indoor with the help of convolutional neural network [195]. In addition, camera-based monitoring can extract multiple vital signs (Table 1). It could be included in multi-camera and multi-person measurements. Future research might focus on these techniques in real-world scenarios to develop more practical applications with sufficient reliability. This can also be developed into optical-sensors-based devices and smartphone-based or webcam-based applications that might results in continuous vital signs monitoring or periodic measurement [196,197]. In addition, in the last 20 years, the world has frequently faced a great number of pandemic situations such as SARS in 2002, swine flu in 2009, Ebola in 2014, and now the coronavirus in 2020. So, the current situation urgently requires developing towards noncontact and unobtrusive vital signs monitoring devices without virus transmission. It might be helpful in the pandemic situation for noncontact monitoring of vital signs and for people under home quarantine. 

Our review demonstrated a wide variety of clinical and nonclinical applications, including smart homes and smart cars, which allow for continuous monitoring of vital signs in everyday life [33,35]. In the smart home, camera-based monitoring of vital signs can be combined with gait analysis, fall detection, and sleep monitoring. This could minimize costs and improve out-of-hospital treatments. To reach these application areas, further developments are required. The technology needs robustness in dynamic settings and be able to cope with movement, facial expression, varying illumination, and different camera-subject distance. Furthermore, it requires the development of robust algorithms [87]. Nonetheless, future research should be directed toward mobile health (mHealth) applications to provide users a timely and consistent feedback of their health status [158]. mHealth must perform without hardware attachment and provide user-friendly interfaces. Due to the increasing number of smartphone users and continuously improved smartphone sensor technology (e.g., camera), we expect that many studies of this review transform into smartphone applications and serve in telemedicine.

## 5. Conclusions

Camera-based techniques monitor vital signs unobtrusively. This review provides an overview of data acquisition technology (hardware), image and signal processing (software), accuracy, and application areas. As of today, HR and RR are reliably monitored using RGB cameras in controlled settings only, but other vital signs are still lacking robust and sufficiently precise systems. Subject, camera, setting, and environmental parameters have a significant impact on the accuracy. To overcome these effects, robust algorithms based on advanced signal processing or DL are urgently needed. Additionally, fusion-based approaches (e.g., multiple ROI or multiple cameras) bear the potential of enhancing reliability. Different ranges of hardware and software parameter can be investigated to obtain the best possible results for various environments. With respect to the COVID-19 pandemic, a potential application could be to deploy cameras for smart-home-based health monitoring of subjects undergoing quarantine. Recently, the U.S. Food and Drug Administration (FDA) approved smart phone applications for therapy e.g., for tinnitus [198]. In light of this development, smartphone camera-based vital sign monitoring could also be part of digital health. We call for collaboration across the world to collect publicly available dataset of large diversity and size as a basis to make camera-based vital signs monitoring sufficiently robust to be translated into smartphone apps and regularly used in future mHealth and telemedicine. 

## Figures and Tables

**Figure 1 sensors-22-04097-f001:**
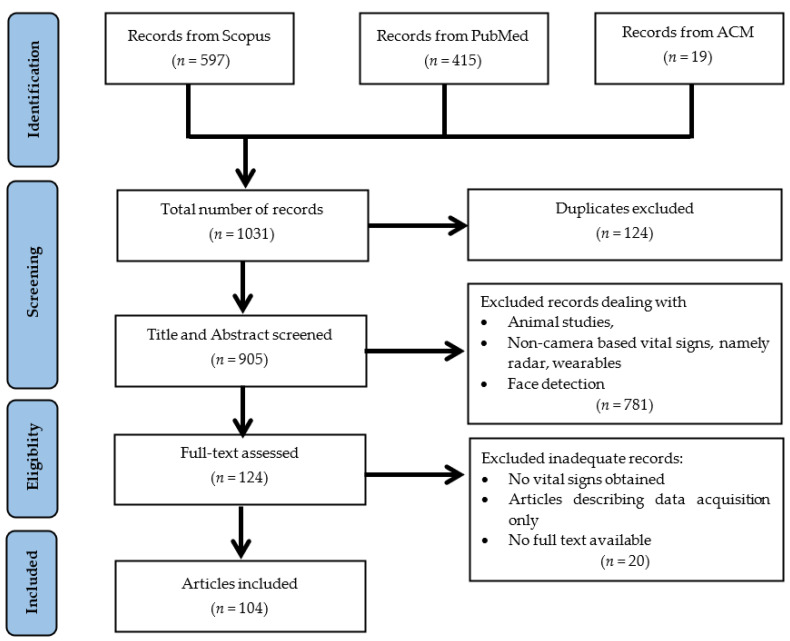
PRISMA diagram of literature screening.

**Figure 2 sensors-22-04097-f002:**
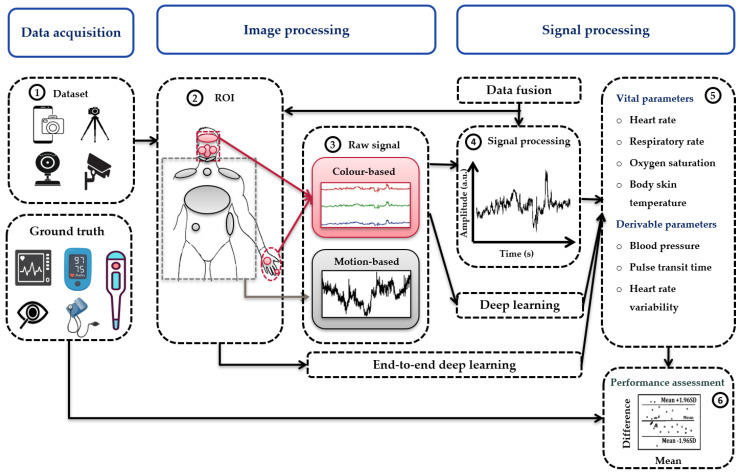
General flow diagram of vital sign measurements from video/image data.

**Figure 3 sensors-22-04097-f003:**
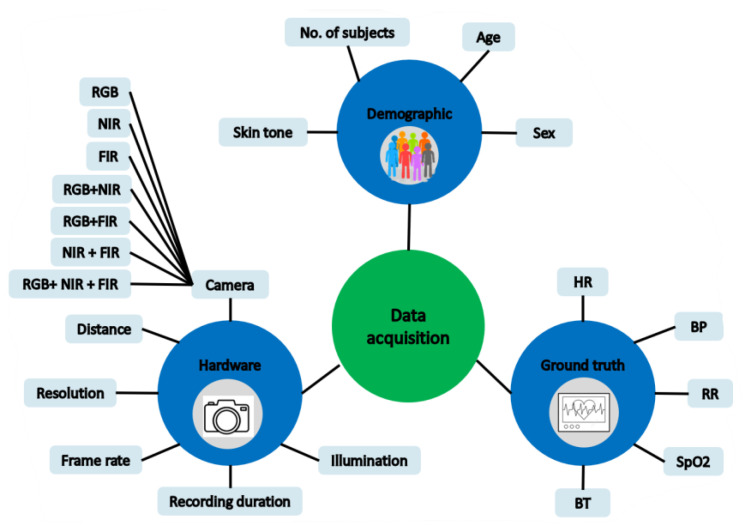
Data acquisition according to study design, hardware, and ground truth.

**Figure 4 sensors-22-04097-f004:**
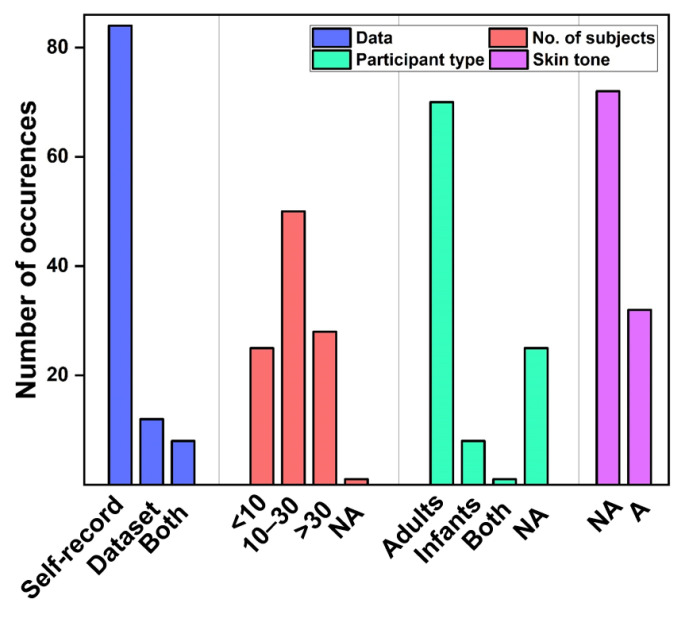
Distribution of subject information including obtained data, number of subjects, participant type and available (A) and not available (NA) info of skin tone.

**Figure 5 sensors-22-04097-f005:**
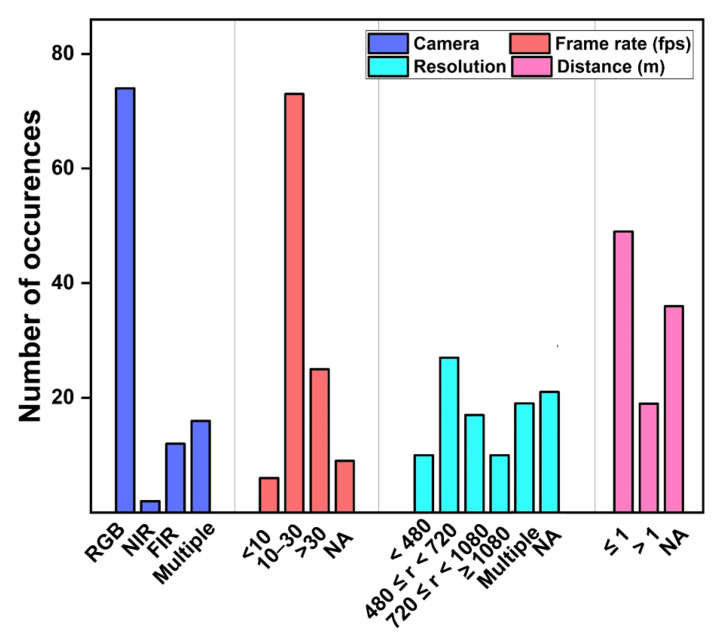
Distribution of hardware parameters, namely camera type, frame rate, resolution (r), and camera-subject distance.

**Figure 6 sensors-22-04097-f006:**
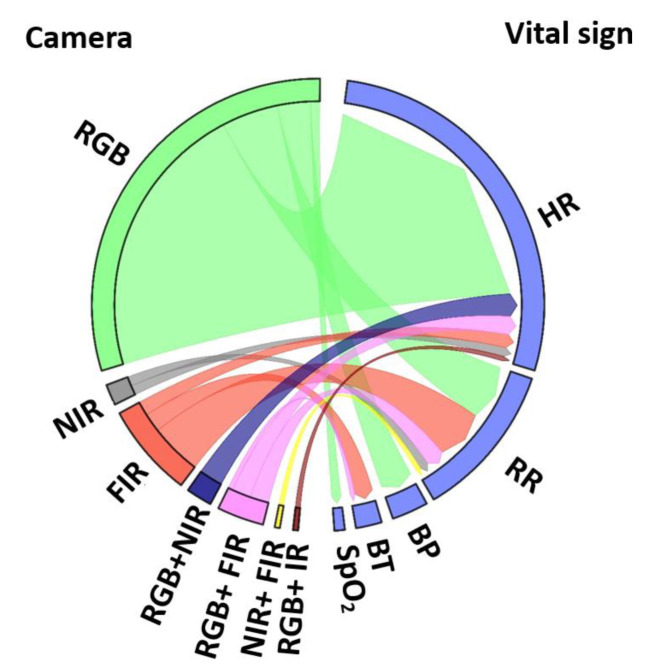
Chord diagram of conceptual semantics between camera and vital sign.

**Figure 7 sensors-22-04097-f007:**
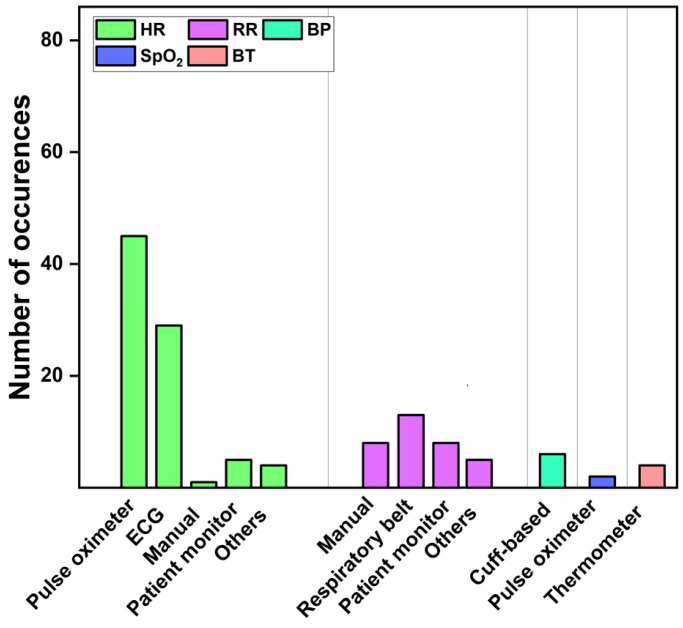
Distribution of ground truth.

**Figure 8 sensors-22-04097-f008:**
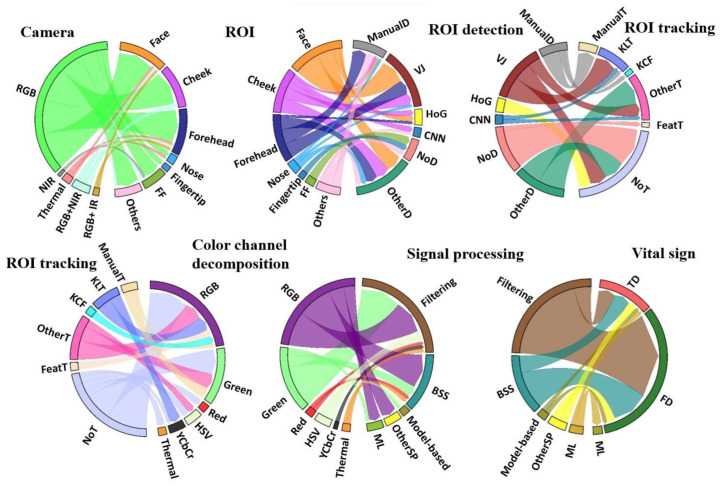
Comprehensive work flow from various camera images to HR vital sign (Note: FF: full frame, VJ: Viola Jones algorithm, HoG: Histogram of oriented gradients; CNN: convolutional neural network; ManualD: manual ROI detection, NoD: no ROI detection, OtherD: Other ROI detection, ManualT: manual ROI tracking, KLT: Kanade–Lucas–Tomasi; KCF: kernel correlation filter; featT: feature tracking, OtherT: other ROI tracking, NoT: no ROI tracking, BSS: blind source separation algorithms, otherSP: other signal processing, ML: Machine learning FD: frequency domain, TD: time domain).

**Figure 9 sensors-22-04097-f009:**
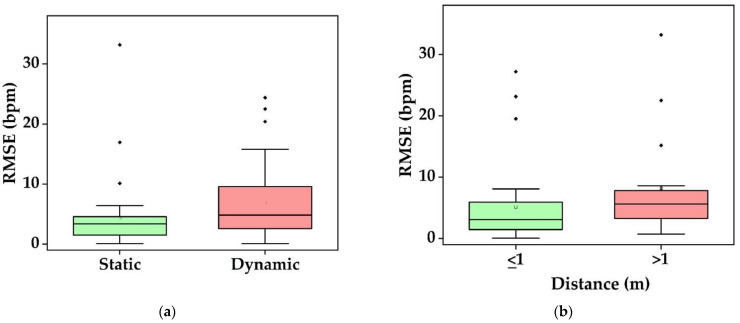
(**a**) RMSE of HR in static and dynamic conditions and, (**b**) varying distance between the subject and camera.

**Figure 10 sensors-22-04097-f010:**
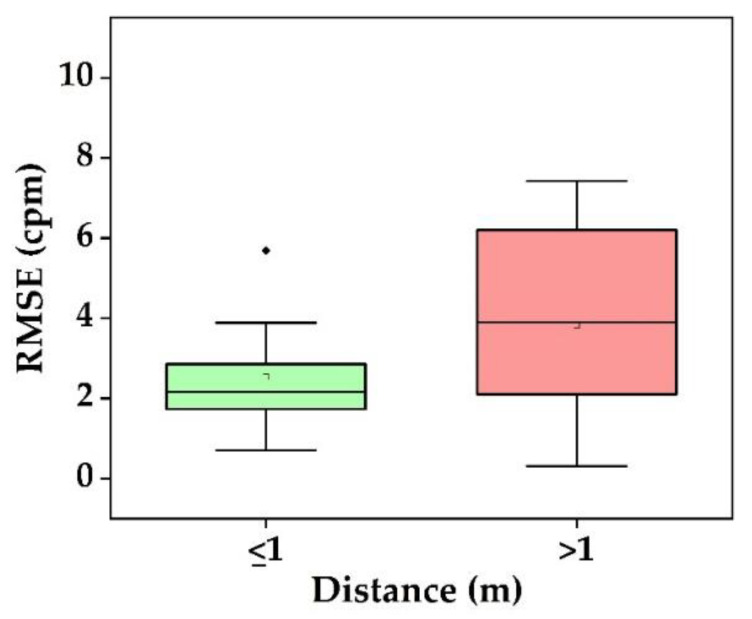
RMSE of RR estimation with varying distance.

**Figure 11 sensors-22-04097-f011:**
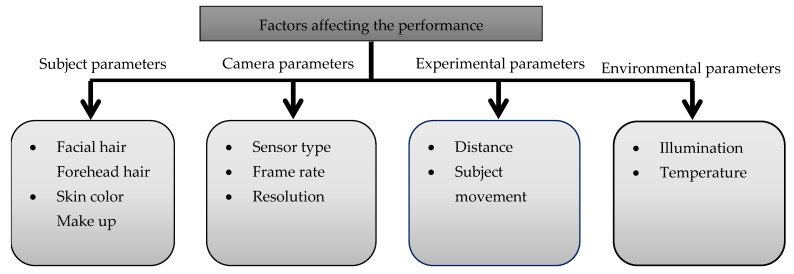
Factors affecting the vital signs performance.

**Figure 12 sensors-22-04097-f012:**
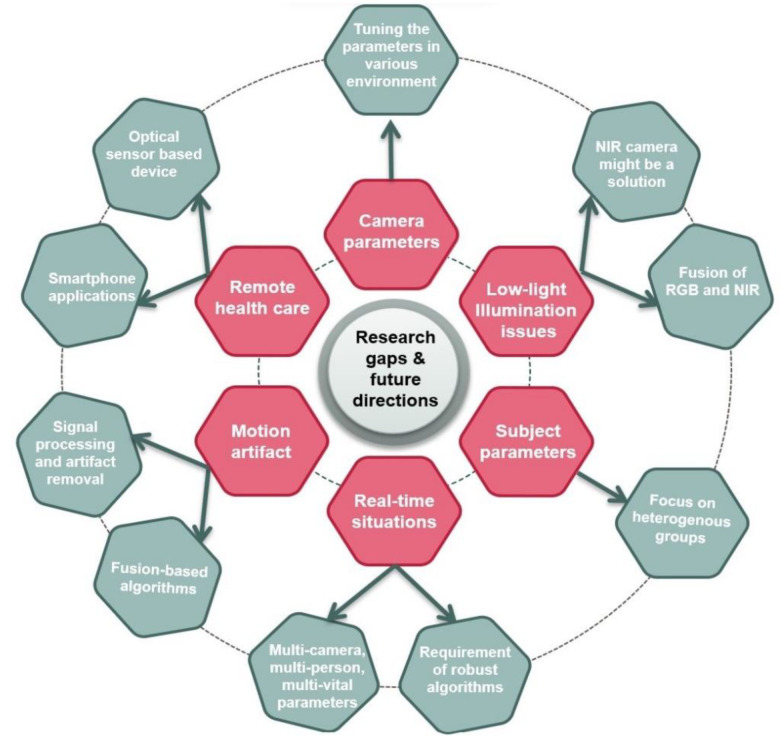
The framework of main research gaps and future directions. In this, red color boxes represent the research gaps, and green color boxes refer to future research directions.

**Table 1 sensors-22-04097-t001:** Number n of articles from the considered years: January 2018 to April 2021.

Article Year	HR	RR	BP	BST	SpO_2_	Multiple	Total
2018	25	2	1	1	0	7	36
2019	14	9	3	0	0	3	29
2020	16	5	2	1	0	3	27
2021	5	4	0	0	1	2	12
Total	60	20	6	2	1	15	104

**Table 2 sensors-22-04097-t002:** Existing databases.

Database	No. of Subjects	Camera Type	Camera Detail	Frame Rate (fps)	Resolution (px × px)	Ground Truth
MAHNOB-HCI [133,139]	27	RGB	Allied Vision Stingray F-046C; F-046B	60	780 × 580	ECG
DEAP [134,140]	22	RGB	Sony DCR-HC27E	50	720 × 576	PPG, Respiration, EEG, EOG, EMG, GSR, BT
FAVIP [83,141]	15	RGB	Samsung galaxy S3 and iPhone 3GS	30	1280 × 720	Pulse oximeter
UBFC-RPPG [135,142,143]	42	RGB	Logitech C920 HD pro	30	640 × 480	Pulse oximeter
PURE [136,144]	10	RGB	evo274CVGE	30	640 × 480	Finger pulse oximeter
Pulse from face [78,145]	13	RGB	Nikon D5300 camera	50	1280 × 720	Two Mio Alpha II wrist heart rate monitors
VIPL_HR [137,146]	107	RGB	Logitech C310	25	960 × 720	CONTEC CMS60C blood volume pulse recorder
		NIR	Realsense F200	30	640 × 480	
		RGB		30	1920 × 1080	
		RGB	HUAWEI P9 smart phone	30	1920 × 1080	
COHFACE [138,147]	40	RGB	Logitech HD C525	20	640 × 480	Blood volume pulse sensor, respiratory belt
MMSE-HR [80,148]	40	RGB, IR	Di3D dynamic imaging system, FLIR A655sc	25	1040 × 1392	Biopac MP150 system—BP, HR
				50	640 × 480	
TokyoTech Remote PPG [118,149]	9	RGB, NIR	Prototype RGB-NIR camera	300	640 × 480	Contact PPG sensor
MR-NIRP [41,150]	18	RGB, NIR	FLIR Grasshopper3, Point Grey Grasshopper	30	640 × 640	Finger pulse oximeter

**Table 4 sensors-22-04097-t004:** Preprocessing using different filtering methods.

Filtering	Description
Detrending filter [89,92,99,113,130,153,154,160]	It removes the trend in signal
Moving average filter [29,55,116,125,126,132,154,166]	It smooths a signal and suppresses high frequency noise
Band-pass filter [52,75,80,95,98,109,116,122,126,132,152,167]	It eliminates the frequency components outside the bandwidth range

**Table 5 sensors-22-04097-t005:** Various algorithms for HR extraction.

Signal Processing Techniques	Characterization
ICA [176,177]	It decomposes the signal and extracts independent components of pulse information from temporal RGB traces.
PCA [51,53,175,178]	It utilizes a statistical technique to obtain uncorrelated components from RGB traces [151].
GREEN [38,104,130]	In blood, hemoglobin and oxyhemoglobin absorb light of 520–580 nm, which is in the range of the camera’s green filter [38,104,130]. Hemoglobin absorbs green light in sufficient depth [81]. Therefore, the green channel is reported to have more information compared to blue or red channels [179].
CHROM [162]	A linear combination of chrominance signals with the assumption of skin color necessitates a priori knowledge and eliminates motion artifacts but it may fail if pulse and specular signals are same [59].
POS [116]	It projects the RGB-derived signals onto a plane orthogonal to the temporally normalized skin tone component.
Spatial subspace rotation [162,180]	It utilizes the subspace of skin pixels and rotation measurements for extracting cardiac pulse information but it may require complete continuous sequence of camera frames to recover the pulse wave [120].
Kernel density ICA [59]	It does not require a prior assumption of probability distributions of hidden sources and so-called semi-BSS method.

**Table 6 sensors-22-04097-t006:** Studies using RR by utilizing camera-based approaches.

Ref.	Camera Type	Camera Details	Frame Rate (fps)	Resolution (px × px)	ROI	Ground Truth	Results
[152]	FIR	Infratec VarioCAM HD head	30	1024 × 768	Nose	Philips IntelliVue MP30 monitor	Correlation coefficient (CC): 0.607 upon arrival, 0.849 upon discharge
[29]	FIR	Optrics PI 450	80	382 × 288	Nose	Manual counting	CC: near distance: 0.960; far distance: 0.508;
[27]	FIR	InfraTec VarioCAM HD head	30	1024 × 768	Full frame and split into sub-ROI	Adults: Respiratory belt, Infants: Dräger M540 patient monitor	Root mean square error (RMSE): healthy adults: (sit still: 0.31 ± 0.09 cpm, stimulated breathing: 3.27 ± 0.72 cpm), infants: 4.15 ± 1.44 cpm
[125]	FIR	FLIR SC3000	30	320 × 240	Nostril area and mouth, nose and cheeks enclosed	Subject finger flexion (upward–inhalation, downward–exhalation)	RMSE: 3.40 cpm
[94]	FIR	Optrics PI-450	27	382 × 288	Nose	Manual	RMSE: stay still: 3.81 cpm, moving: 6.20 cpm
[25]	FIR	FLIR Lepton 2.5, FLIR lepton 3.5	8.7	60 × 80; 120 × 160	Full frame	Philips MX700 patient monitor	Mean absolute error (MAE): 2.07 cpm
[32]	FIR	Seek Thermal Compact PRO for iPhone	17	640 × 480	Highest temperature point and around it	Respiration belt	RMSE: 1.82 ± 0.75 cpm
[128]	FIR	FLIR T450sc	30	-	Nose	GE healthcare patient monitor, visual inspection	CC: 0.95
[57]	FIR	Infratec ImageIR 9300	50	1024 × 768	Nose	piezo plethysmography, IntelliVue MP70 patient monitor	RMSE: 0.71 ± 0.30 cpm
[31]	FIR	FLIR T-420	10	320 × 240	Nostril	Respiratory volume monitor	CC: 0.86 before sedation
[131]	RGB; FIR	Dual camera DFK23U618; FLIR A315	15	640 × 480; 320 × 240	Nose and mouth	Respiration effort belt	CC: 0.87; RMSE: 1.73 cpm
[106]	RGB; FIR	Dual camera DFK23U618; FLIR A315	15	640 × 480; 320 × 240	Nasal area	Respiratory effort belt	RMSE: standing: 1.44 cpm, seated position with body movement: 2.52 cpm
[166]	RGB, FIR	MAG62 thermal imager	10	640 × 480	Nostril region	Sleep respiratory monitor	Coefficient of determination: 0.905
[60]	NIR; FIR	NIR: see3cam_CU40, FIR: FLIR lepton version 3.5	15; 8.7	336 × 190, 160 × 120	Chest, Nostril	Respiratory belt	RMSE: 4.44 cpm
[26]	RGB; FIR	RGB: IDS UI-2220SE; FIR: FLIR Lepton 2.5	20; 8.7	576 × 768; 60 × 80	Full frame	Philips patient monitor	MAE: 5.36 cpm
[124]	RGB	Point Grey Flea 3 GigE	-	648 × 488	Chest	Polysomnography	Mean error: non magnified: 0.874 cpm; magnified: 0.67 cpm
[24]	RGB	IP camera	10	320 × 180	Full frame	ECG impedance pneumography	CC: 0.948; RMSE: 6.36 cpm
[184]	RGB	IDS uEye-2220	20	--	Torso	Capnography	All clothing styles and respiratory patters (CC: 0.90–1.00) except winter coat-slow-deep scenario (CC:0.84)
[108]	RGB	Digital camera	24/30	1920 × 1080 /1280 × 720	Abdominal area	Dräguer NICU monitor	CC: 0.86
[91]	RGB	IDS UI-3160CP	120	1920 × 1080	Face	Upper chest signal	Error: −0.25 to 0.5 cpm
[53]	NIR	Point Grey Firefly MV USB 2.0	30	640 × 480	Full frame	PSG, ECG, Inductance plethysmography	CC: 0.80; RMSE: 2.10 ± 1.64 cpm; MAE: 0.82 ± 0.89 cpm
[102]	RGB	Nikon D610, D5300	30	1920 × 1080	Abdominal area	Philips intellivue monitor	Limits of agreement: −22 to 23.6 cpm
[105]	RGB	CCD camera	30	1280 × 720	Jugular notch	Differential digital pressure sensor	MAE: 0.39 cpm; Limits of agreements: (slim fit: ±0.98 cpm, loose fit: ±1.07 cpm)
[43]	RGB	Smartphone LG G2	30	-	Forehead	Visual inspection, Pulse oximeter, Heart rate monitor	RMSE: hue: 3.88 cpm; Green: 5.68 cpm
[89]	RGB	Logitech C922/ GigE Sony XCG-C30C	60	1280 × 720/ 659 × 494	Forehead, nose, cheeks	Respiratory belt	Relative error < 2% and inter quartile range < 5%
[51]	NIR	Monochromatic infrared camera	62	640 × 240	Neck area with chin and upper chest	Chest belt	CC: 0.99; RMSE: 0.70 cpm
[99]	RGB	JAI 3-CCD AT-200CL	20	1620 × 1236	Skin	Philips patient monitor	MAE: 3.5 cpm
[52]	NIR	Thermal imager MAG62, Avigilon H4 HD Dome	-	640 × 480	Chest	Manual	Coefficient of determination: dataset 1: 0.92, dataset 2: 0.87
[185]	RGB	Smartphone Galaxy S9+	240	1920 × 1080	Abdomen or waist area	Manual counting	Accuracy 99.09%
[56]	RGB	Canon camera	-	-	Cheeks enclosed	PPG sensor, Respiratory belt	RMSE: 2.16 cpm

## Data Availability

Source data for all figures are provided with the paper and Appendix A.

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
