# Peer review of "Continuous Monitoring of Vital Signs Using Cameras: A Systematic Review"

_sensors, 2022, doi:10.3390/s22114097_

Round 1

Reviewer 1 Report

The manuscript of Selvaraju et al. aimed to summarize the knowledge regarding the camera-based vital sign monitoring. The topic is interesting, however I do have some concerns:

Line 56-57: SpO2 may also refer to the amount of oxygen carried by blood, but it is not always true. Precisely, it is the ratio of oxygenated hemoglobin to total hemoglobin. This parameter points to hemoglobin oxygenation - thus the mention that it is essential physiological parameter to assess the blood flow (line 57-58) is not fully precise.

ECG is mentioned several times - but can it be conducted in a non-contact way? The references to the ECG should be better clarified.

line 61 - “contact-based measures are obtrusive, harmful, and may result in skin irritations” - really harmful? - this expression is too strong

Line 68 - pulse wave refers to dilation of the arterial system generated by cardiac systole - it does not refer to blood flow or perfusion - as it is written in the manuscript.

Line 70 - Instead of “These pulsatile component of the pulse wave in the capillary vessels causes” I recommend following statement: “These pulsatile blood flow causes…”. In the capillaries, blood flow is continuous, not pulsatile.

Line 74-75 - The use of image or video data offers noninvasive and unobtrusive measurement of vital signs. - this is the aim, but is it already true?

Line 118 - BST - should be defined earlier in the first use

Line 135 - vascular status? What exactly to understand?

Figure 2 - I recommend to add detailed description of this figure. Or indicate its parts to guide the readers better.

What is the importance of presented gender distribution (tbl 3)? I think it is not relevant in the proposed article at all.

Tbl 5-7 - significant portion of reviewed articles with not reported information - it may be discussed in form of recommendations for future studies.

In the tbl. 8, obtained clinical parameters by individual studies could be also added (like ground truth).

links to websites (page 9-10) - into references?

Some abbreviations are not necessary - KLT (line 329), KCF (line 330), EVM (line 341), AAMI (line 380).

If possible, I recommend to reduce the count of abbreviations.

Line 385 - infants - what age category?

Line 388 - RR - basic physiological knowledge - you can add reference to a textbook

In the legends, all abbreviations should be explained

Chapter 3.5 - it represents only summary of different approaches without additional comment, recommendation, conclusion…

Such conclusions: “increasing the illumination reduces the error” or “increased melanin concentration lowers the performance of HR estimation” were expected and well confirmed - any suggestion to problem solution? This could increase the novelty of the article as well as its contribution.

Author Response

Response to Reviewer 1

General comments:

The manuscript of Selvaraju et al. aimed to summarize the knowledge regarding the camera-based vital sign monitoring. The topic is interesting; however, I do have some concerns:

General response:

Many thanks for the comments and the valuable suggestions.

In the following sections, we will address the identified issues point-by-point. Responses are in black text.

Specific comments:

Point 1:

Line 56-57: SpO2 may also refer to the amount of oxygen carried by blood, but it is not always true. Precisely, it is the ratio of oxygenated hemoglobin to total hemoglobin. This parameter points to hemoglobin oxygenation - thus the mention that it is essential physiological parameter to assess the blood flow (line 57-58) is not fully precise.

Response 1:

Thanks for pointing out the unclear statement in introduction section. We update it to clarify the statement.

The introduction is now read alike this, “SpO2 measures the relative concentration of oxygenated hemoglobin with respect to the total amount of hemoglobin, and it is an essential physiological parameter to assess the oxygen supply to the human body”. (Line 57-60)

Point 2:

ECG is mentioned several times - but can it be conducted in a non-contact way? The references to the ECG should be better clarified.

Response 2:

Thank you for your comment. The non-contact method of measuring ECG was included to the revised manuscript.

The introduction is now look alike this, “non-contact and unobtrusive vital sign monitoring is an emerging field of research. For example, capacitive coupled ECG can be used to measure the electrical activity of the heart in an indirect way. It is recorded without physical contact to the skin through clothing and can be integrated into chairs, beds, or driver seats [15]. However, capacitive ECG is strongly affected by movement artifacts, coupling impedance fluctuation, and electromagnetic interference [16]”. (Line 65-70)

Point 3:

line 61 - “contact-based measures are obtrusive, harmful, and may result in skin irritations” - really harmful? - this expression is too strong

Response 3:

We update it to clarify the statement.

The introduction is now read alike this, “However, contact-based measures are obtrusive and may result in skin irritations. For instance, wet ECG electrodes usually need replacement after 48 hours.” (Line 63-65).

Point 4:

Line 68 - pulse wave refers to dilation of the arterial system generated by cardiac systole - it does not refer to blood flow or perfusion - as it is written in the manuscript.

Response 4:

We update it to clarify the statement.

The introduction is now read alike this, “Due to the contraction and relaxation phases of the human heart, the blood travels throughout the vascular system.” (Line 75-77).

Point 5:

Line 70 - Instead of “These pulsatile component of the pulse wave in the capillary vessels causes” I recommend following statement: “These pulsatile blood flow causes…”. In the capillaries, blood flow is continuous, not pulsatile.

Response 5:

Thank you for providing your suggestion. We update the statement in the revised manuscript.

The introduction is now read alike this, “These pulsatile blood flow causes subtle changes in the skin color that is detected by image recording devices”. (Line 78-80).

Point 6:

Line 74-75 - The use of image or video data offers noninvasive and unobtrusive measurement of vital signs. - this is the aim, but is it already true?

Response 6:

Thank you for providing your comment. Camera based vital sign measurements have recently attracted increasing scientific attention and can avoid the complications associated with contact-based monitors such as, tethering, skin irritation and loss of contact. It might offer noninvasive and unobtrusive measurement of vital signs. It is still being investigated. Hence, we update the statement as following: “The use of image or video data might offer noninvasive and unobtrusive measurement of vital signs”. (Line 82-83).

Point 7:

Line 118 - BST - should be defined earlier in the first use

Response 7:

Thank you. We have updated the BST definition.

Introduction section is now read alike this, “body skin temperature (BST) is a non-contact and non-invasive method of measuring the temperature continuously by the detection of infrared-radiation emitted from the body [58]. It can be monitored using FIR. FIR measures this infrared radiation which results in estimating BST”. (Line 124-128).

Point 8:

Line 135 - vascular status? What exactly to understand?

Response 8:

Vascular status provides the information regarding blood vessels health like narrowed or blocked blood vessel. It might reflect in blood wave velocity parameter and pulse transit time.

The text reads now: “Zaunseder et al. focused on HR and HRV, as well as other derivable parameters including pulse transit time and pulse wave velocity to remotely examine the peripheral vascular system [66].” (Line 143-145)

Point 9:

Figure 2 - I recommend to add detailed description of this figure. Or indicate its parts to guide the readers better.

Response 9:

Thank you. Detailed description is added and Figure 2 is updated in the revised manuscript.

Added description can be seen in results section, “At first, video or image data from smartphones, webcams, or digital cameras is acquired with synchronously obtained ground truth (Sec. 3.1). Existing datasets are also used (Sec. 3.2). As shown in section 3.3, image processing techniques are then employed to detect the region of interest (ROI) and the raw signal is extracted from the ROI. In section 3.4, we dis-cuss signal processing techniques, mainly deep learning (DL) approaches, to remove motion and illumination artifacts. After optional data fusion (Sect. 3.5), the vital sign is extracted from the signal (Sect. 3.6) and compared to the ground truth using performance metrics (Sect. 3.7).” (Line 218-225)

Point 10:

What is the importance of presented gender distribution (tbl 3)? I think it is not relevant in the proposed article at all.

Response 10:

Thank you. Majority of the studies explored on unbiased gender distribution or only male or female distribution. There is no study which have extensively presented on gender distribution based iPPG measurement, as per our knowledge. There might be an impact on vital sign based on gender subsets. In the future, data with equal gender distribution can be obtained to investigate on the influence of performance. It is mentioned in discussion section along with age distribution.

The discussion is now read alike this, “In addition, there might be bias in age and gender due to changes in the skin proper-ties. There is no work that focus exclusively on healthy elder subjects, children above age of two, or a particular gender. Hence, further studies are required to effectively evaluate the influence of age and gender.” (Line 764-767)

Point 11:

Tbl 5-7 - significant portion of reviewed articles with not reported information - it may be discussed in form of recommendations for future studies.

Response 11:

Thank you for your comments. We have updated the discussion section in the revised manuscript.

The discussion section is now read alike this, “Future research is required to focus on resolving the issues by tuning these parameters, namely camera-subject distance, camera parameters, and illumination, in various in-door and outdoor environments to obtain the best possible results that might improve the robustness in practical applications (Fig. 12). As of today, subject and camera pa-rameters are not mentioned in most of the article. Researchers need to improve and follow existing guidelines such as STARE-HI [194].” (Line 774-780)

[194] Talmon, J.; Ammenwerth, E.; Brender, J.; Dekeizer, N.; Nykanen, P.; Rigby, M. STARE-HI—Statement on Reporting of Evaluation Studies in Health Informatics. International Journal of Medical Informatics 2009, 78, 1–9, doi:10.1016/j.ijmedinf.2008.09.002.

Point 12:

In the tbl. 8, obtained clinical parameters by individual studies could be also added (like ground truth).

Response 12:

Thank you for your suggestion. We have updated the ground truth information in tbl. 2, can be seen below:

Table 2. Existing databases.

Database

No. of       subjects

Camera type

Camera detail

Frame rate

(fps)

Resolution

(px x px)

Ground truth

MAHNOB-HCI

[132,133]

27

RGB

Allied Vision Stingray

F-046C; F-046B

60

780 x 580

ECG

DEAP

[134,135]

22

RGB

Sony DCR-HC27E

50

720 x 576

PPG, Respiration, EEG, EOG, EMG, GSR, BT

FAVIP

[81,136]

15

RGB

Samsung galaxy S3 and

iphone 3GS

30

1280 x 720

Pulse oximeter

UBFC-RPPG

[137–139]

42

RGB

Logitech C920 HD pro

30

640 x 480

Pulse oximeter

PURE

[138,140,141]

10

RGB

evo274CVGE

30

640 x 480

Finger pulse oximeter

Pulse from face (PFF)

[76,142]

13

RGB

Nikon D5300 camera

50

1280 x 720

Two Mio Alpha II wrist heart rate monitors

VIPL_HR

[143,144]

107

RGB

Logitech C310

25

960 x 720

CONTEC CMS60C blood volume pulse sensor

NIR

RGB

Realsense F200

30

30

640 x 480

1920 x 1080

RGB

HUAWEI P9 smart phone

30

1920 x 1080

COHFACE

[145,146]

40

RGB

Logitech HD C525

20

640 x 480

Blood volume pulse sensor, respiratory belt

MMSE-HR

[78,147]

40

RGB, IR

Di3D dynamic imaging system, FLIR A655sc

25,

50

1040 x 1392,
640 x 480

Biopac MP150 system - BP, HR

TokyoTech Remote PPG [117,148]

9

RGB, NIR

Prototype RGB-NIR camera

300

640 x 480

Contact PPG sensor

MR-NIRP

[38,149]

18

RGB, NIR

FLIR Grasshopper3, Point Grey Grasshopper

30

640 x 480

Pulse oximeter

Point 13:

links to websites (page 9-10) - into references?

Response 13:

We have updated the websites link into references. Please see reference section (Ref. 135, 137, 138, 141,143,145,146,148,149,150,151)

Point 14:

Some abbreviations are not necessary - KLT (line 329), KCF (line 330), EVM (line 341), AAMI (line 380). If possible, I recommend to reduce the count of abbreviations.

Response 14:

As suggested, we have removed the abbreviations and reduced the abbreviations count.

Point 16:

Line 385 - infants - what age category?

Response 16:

Thank you for your clarification. Infants with less than one year age group have a HR of 110-160 breaths per minutes.

Introduction section is now read alike this, “Since infants under the age of one year have a higher HR than adults, ranging from 110 to 160 bpm [98,170], research on infants reported frequencies of 1.83-2.67 Hz [109], 1.3-5 Hz [110], and 1.67-3.33 Hz [101].” (Line 401-403).

  • Nageotte, M.P. Fetal Heart Rate Monitoring. Seminars in Fetal and Neonatal Medicine 2015, 20, 144–148, doi:10.1016/j.siny.2015.02.002.

Point 17:

Line 388 - RR - basic physiological knowledge - you can add reference to a textbook

Response 17:

As suggested, we have added the textbook reference for RR.

Introduction section is now read alike this, “The RR of adult lies between 12 cpm and 20 cpm [171,172].” (Line 404-405).

  • Mader, S. S.; Understanding Human Anatomy & Physiology; 5th ed.; WCB/McGraw-Hill, 2004;
  • Hill, B.; Annesley, S.H. Monitoring Respiratory Rate in Adults. British Journal of Nursing 2020, 29, 4.

Point 18:

In the legends, all abbreviations should be explained

Response 18:

Thank you. We have explained all abbreviations.

Point 19:

Chapter 3.5 - it represents only summary of different approaches without additional comment, recommendation, conclusion…

Response 19:

The chapter 3.5 were extended with additional recommendation and conclusion.

Particularly, added description, “In summary, fusion-based techniques can significantly improve the robustness of vital sign monitoring, despite the motion and illumination problems in real-time situations [15]. For instance, sensor fusion techniques that employ multiple cameras positioned at different angles can reliably track the ROI of the subject and compensate motion artifacts (e.g., head turn). A combination of RGB and NIR cameras for day and night environments is helpful in low light applications [187]. Furthermore, signal fusion might allow single camera approaches to assess multiple ROIs simultaneously.” (Line 539-545).  

Point 20:

Such conclusions: “increasing the illumination reduces the error” or “increased melanin concentration lowers the performance of HR estimation” were expected and well confirmed - any suggestion to problem solution? This could increase the novelty of the article as well as its contribution.

Response 20:

Thank you for your comment. We have discussed these comments in the revised manuscript.

The results section is now read alike this:

“HR measurement using RGB cameras suffer from darkness, whereas NIR cameras yield better results” (Line 587-588).

“A combination of RGB and NIR cameras for day and night environments is helpful in low light applications [187].” (Line 543-544)

Discussion section is now read alike this, “RGB images with proper illumination provides better signal performance, whereas NIR camera have a lower SNR signal due to the lower absorption of hemoglobin [41]. However, a few studies recently concentrated on NIR camera-based vital sign monitoring owing to the advantages of functioning in dark conditions (e.g., sleep monitoring, driver monitoring) [34,41,93]. Even so, it is not explored widely. Future research on NIR cameras might yield a path for camera-based vital sign monitoring in all light conditions.” (Line 753-759)

“Most of the studies disregard skin colors although the melanin concentration has a strong impact on the measurements. Developing robust algorithms using DL might help here.” (Line 767-769)

Reviewer 2 Report

The paper presents a systematic review of continuous camera-based vital sign monitoring using Scopus, PubMed, and the Association for Computing Machinery (ACM) databases. It analyzed articles published between January 2018 and April 2021 in the English language. It retrieved 905 articles and screened them regarding title, abstract, and full text, and 104 scientific papers remained.

The paper is well organized, and the length is appropriate. The title is chosen correctly, and the abstract provides sufficient information to give a clear idea of what to expect from the paper.

The study methods are appropriate, and the data are valid

The results are well highlighted, and the conclusions are adequate.

The references are relevant and correctly chosen, and related work is discussed and cited appropriately.

The technical depth of the paper meets the requirements for a scientific article published in a quality journal.

Author Response

Response to Reviewer 2

General comments:

The paper presents a systematic review of continuous camera-based vital sign monitoring using Scopus, PubMed, and the Association for Computing Machinery (ACM) databases. It analyzed articles published between January 2018 and April 2021 in the English language. It retrieved 905 articles and screened them regarding title, abstract, and full text, and 104 scientific papers remained.

The paper is well organized, and the length is appropriate. The title is chosen correctly, and the abstract provides sufficient information to give a clear idea of what to expect from the paper.

The study methods are appropriate, and the data are valid

The results are well highlighted, and the conclusions are adequate.

The references are relevant and correctly chosen, and related work is discussed and cited appropriately.

The technical depth of the paper meets the requirements for a scientific article published in a quality journal.

General response:

Many thanks for the positive words.

Reviewer 3 Report

This review discusses data acquisition technology (hardware), image and signal processing technology (software), accuracy, and application areas. HR and RR are currently monitored reliably using RGB cameras in controlled environments, but other vital signs continue to lack robust and precise systems. The accuracy is highly dependent on the subject, camera, setting, and environmental parameters. However, I believe that revision is necessary to maintain the Journal's high publishing standards. 
1. The Abstract and Conclusion sections are inadequate. The majority of the contents are covered in these sections through access to content and discussion of collaborative works. The authors should rewrite both sections and focus exclusively on the research findings. Not the method of obtaining the papers, but the method of writing the article. 
2. The heading "2. Materials and Methods" is incorrect. This section contains no "materials." 
The entire manuscript appears to be a book chapter, and readers are missing "novelty" points. Readers are no longer interested in learning "how many" pieces of literature are available in various databases. After careful consideration, I have decided to reject this work because it is only marginally incremental in comparison to the state-of-the-art literature and does not meet the high quality standards for publication in the Sensors Journal.

Author Response

Response to Reviewer 3

General comments:

This review discusses data acquisition technology (hardware), image and signal processing technology (software), accuracy, and application areas. HR and RR are currently monitored reliably using RGB cameras in controlled environments, but other vital signs continue to lack robust and precise systems. The accuracy is highly dependent on the subject, camera, setting, and environmental parameters. However, I believe that revision is necessary to maintain the Journal's high publishing standards. 

General response:

Thank you for your comments. We have updated the manuscript based on your comments.

In the following sections, we will address the identified issues point-by-point. Responses are in black text.

Specific comments:

Point 1:

The Abstract and Conclusion sections are inadequate. The majority of the contents are covered in these sections through access to content and discussion of collaborative works. The authors should rewrite both sections and focus exclusively on the research findings. Not the method of obtaining the papers, but the method of writing the article. 

Response 1:

Many thanks for the valuable comment. We have updated the manuscript abstract and conclusion section.

Abstract section is now read alike this: “In recent years, non-contact measurements of vital signs using cameras received a lot of interest. However, some questions are unanswered: (i) which vital sign is monitored using what type of camera, (ii) what is the performance and which factors affect it, (iii) which health issues are addressed by camera-based techniques? Following the preferred reporting items for systematic re-views and meta-analyses (PRISMA) statement, we conduct a systematic review of continuous camera-based vital sign monitoring using Scopus, PubMed, and the Association for Computing Machinery (ACM) databases. We consider articles that were published between January 2018 and April 2021 in English language. We include five vital signs: heart rate (HR), respiratory rate (RR), blood pressure (BP), body skin temperature (BST), and oxygen saturation (SpO2). In total, we retrieve 905 articles and screened them regarding title, abstract, and full text. One hundred and four articles remained: 60, 20, 6, 2, and 1 of the articles focus on HR, RR, BP, BST, and SpO2, respectively, and 15 on multiple vital signs. HR and RR can be measured in red, green, and blue (RGB) and near infrared (NIR) as well as far-infrared (FIR) cameras. So far, BP and SpO2 are monitored with RGB cameras only, whereas BST is derived from FIR cameras only. Under ideal conditions, the root mean squared error is 2.6 bpm, 2.22 cpm, 4.16 mm Hg, 2.85 mm Hg, and 0.86 °C for HR, RR, systolic BP, diastolic BP, and BST, respectively. The estimated error for SpO2 is less than 1 %, but it increases with movements of the subject and the camera-subject distance. Camera-based remote monitoring mainly explores intensive care, post-anaesthesia care, sleep monitoring, but also special diseases such as heart failure. The monitored targets are newborn and pediatric patients, geriatric patients, athletes (e.g., exercising, cycling), and vehicle drivers. Camera-based techniques monitor HR, RR, and BST in static conditions within acceptable ranges for certain applications. In addition, BP and SpO2 are reported in a small number of articles. The research gaps are large and heterogeneous populations, real-time scenarios, moving subjects, and accuracy of BP and SpO2 monitoring.” (Line 14-37)

Conclusion section is now read alike this: “Camera-based techniques monitor vital signs unobtrusively. This review provides an overview of data acquisition technology (hardware), image and signal processing (software), accuracy, and application areas. As of today, HR and RR are reliably monitored using RGB cameras only in controlled settings only, but other vitals still are lacking robust and sufficiently precise systems. Subject, camera, setting, and environmental parameters have a significant impact on the accuracy. To overcome these effects, robust algorithms based on advanced signal processing or DL are urgently needed. Additionally, fusion-based approaches (e.g., multiple ROI or multiple cameras) bear the potential of enhancing reliability. Different ranges of hardware and software parameter can be investigated to obtain the best possible results for various environments. With respect to the COVID-19 pandemic, a potential application could be to deploy cameras for smart home-based health monitoring of subjects undergoing quarantine. Recently, the U.S. Food and Drug Administration (FDA) approved smart phone applications for therapy e.g. for tinnitus [198]. In light of this development, smartphone camera-based vital signs monitoring could also be part of digital health. We call for collaboration across the world to collect publicly available dataset of large diversity and size as a basis to make camera-based vital signs monitoring sufficiently robust to be translated into smartphone apps and regularly used in future mHealth and telemedicine.”  (Line 834-851)

Point 2:

The heading "2. Materials and Methods" is incorrect. This section contains no "materials." 
The entire manuscript appears to be a book chapter, and readers are missing "novelty" points. Readers are no longer interested in learning "how many" pieces of literature are available in various databases. After careful consideration, I have decided to reject this work because it is only marginally incremental in comparison to the state-of-the-art literature and does not meet the high-quality standards for publication in the Sensors Journal.

Response 2:

Thank you for providing your comment. Section 2 is now updated as Methods section. In addition, Novelty points are discussed below:

Most of the existing review article focus on one particular vital sign or disregard how the vital signs are extracted from the camera frames [Hassan et al., 2017; Sikdar et al., 2016; Rouast et al., 2018; Wang et al., 2018; Zaunseder et al., 2018; Steinman et al., 2021]. Other review articles are now out-of-date [Harford et al., 2019; Khanam et al., 2019; Antink et al., 2019]. We present a complete and systematic review of recent advances in camera-based techniques including all five vital signs. By nature, this results in “104” within the short span of time from January 2018 to April 2021. However, we completely agree that “novelty” is also important for the reader, and if the “104” results in small numbers, we now explicitly point this out – as you can see from the detailed answers to the concerns of the other reviewers. We have identified challenges of existing works, got insight in how to address these challenges, and suggested future research prospects.

  • Hassan, M.A.; Malik, A.S.; Fofi, D.; Saad, N.; Karasfi, B.; Ali, Y.S.; Meriaudeau, F. Heart Rate Estimation Using Facial Video: A Review. Biomedical Signal Processing and Control 2017, 38, 346–360, doi:10.1016/j.bspc.2017.07.004.
  • Sikdar, A.; Behera, S.K.; Dogra, D.P. Computer-Vision-Guided Human Pulse Rate Estimation: A Review. IEEE Rev. Biomed. Eng. 2016, 9, 91–105, doi:10.1109/RBME.2016.2551778.
  • Rouast, P.V.; Adam, M.T.P.; Chiong, R.; Cornforth, D.; Lux, E. Remote Heart Rate Measurement Using Low-Cost RGB Face Video: A Technical Literature Review. Comput. Sci. 2018, 12, 858–872, doi:10.1007/s11704-016-6243-6.
  • Wang, C.; Pun, T.; Chanel, G. A Comparative Survey of Methods for Remote Heart Rate Detection From Frontal Face Videos. Bioeng. Biotechnol. 2018, 6, 33, doi:10.3389/fbioe.2018.00033.
  • Zaunseder, S.; Trumpp, A.; Wedekind, D.; Malberg, H. Cardiovascular Assessment by Imaging Photoplethysmography – a Review. Biomedical Engineering / Biomedizinische Technik 2018, 63, 617–634, doi:10.1515/bmt-2017-0119.
  • Khanam; Al-Naji; Chahl Remote Monitoring of Vital Signs in Diverse Non-Clinical and Clinical Scenarios Using Computer Vision Systems: A Review. Applied Sciences 2019, 9, 4474, doi:10.3390/app9204474.
  • Steinman, J.; Barszczyk, A.; Sun, H.-S.; Lee, K.; Feng, Z.-P. Smartphones and Video Cameras: Future Methods for Blood Pressure Measurement. Digit. Health 2021, 3, 770096, doi:10.3389/fdgth.2021.770096.
  • Harford, M.; Catherall, J.; Gerry, S.; Young, J.; Watkinson, P. Availability and Performance of Image-Based, Non-Contact Methods of Monitoring Heart Rate, Blood Pressure, Respiratory Rate, and Oxygen Saturation: A Systematic Review. Meas. 2019, 40, 06TR01, doi:10.1088/1361-6579/ab1f1d.
  • Antink, C.H.; Lyra, S.; Paul, M.; Yu, X.; Leonhardt, S. A Broader Look: Camera-Based Vital Sign Estimation across the Spectrum. Yearb Med Inform 2019, 28, 102–114, doi:10.1055/s-0039-1677914.

Reviewer 4 Report

see attached file

Author Response

Response to Reviewer 4

General comments:

This review provides an overview of data acquisition technology (hardware), image and signal processing (software), accuracy, and application areas. This work is interesting, but several key issues should be solved.

General response:

Thank you for providing your comments and the valuable suggestion.

In the following sections, we will address the identified issues point-by-point. Responses are in black text.

Specific comments:

Point 1:

In Sec.1: for monitoring, devices like doi: 10.1109/TBME.2014.2309951 could be included.

Response 1:

Thank you for your suggested paper. We have included the article in the revised manuscript.

The introduction section is now read alike this, “non-contact and unobtrusive vital sign monitoring is an emerging field of research [14]. For example, capacitive coupled ECG can be used to measure the electrical activi-ty of the heart in an indirect way. It is recorded without physical contact to the skin through clothing and can be integrated into chairs, beds, or driver seats” (Line 65-69)

[14] Zheng, Y.-L.; Ding, X.-R.; Poon, C.C.Y.; Lo, B.P.L.; Zhang, H.; Zhou, X.-L.; Yang, G.-Z.; Zhao, N.; Zhang, Y.-T. Unobtrusive Sensing and Wearable Devices for Health Informatics. IEEE Trans. Biomed. Eng. 2014, 61, 1538–1554, doi:10.1109/TBME.2014.2309951.

Point 2:

Technical impact: for Section “2. Materials and Methods”, what the logical relationship between the following three sub-sections is could be presented by adding one schematic flow diagram;

Response 2:

We noticed that all of the articles can be able to fit in these three stages: 1. Data acquisition for collecting the video/image data and ground truth; 2. image processing techniques to extract the region of interest; 3. Signal processing techniques to remove motion or illuminance variations and estimate the vital sign. A few articles are observed to have end-to-end deep neural networks. Schematic flow diagram is now updated with all the physiological parameters and explained in detail.

Figure 2. General flow diagram of vital sign measurements from video/image data.

Point 3:

in page 7, for the three tables, the presentation style is too complicated for categorized, should be improved, like for Table 2, the x-axis could be the parameter of “Number n of subjects”, and the the y-axis for the purpose of what would be compared as expected;

similar revision should be implemented for the following three tables in page 8;

Response 3:

Thank you for your comment. We have replaced the table into Figure as following:

Figure 4. Distribution of subject information including obtained data, number of subjects, participant type and available (A) and not available (NA) info of skin tone

Figure 5. Distribution of hardware parameters, namely camera type, frame rate, resolution (r), and distance between camera and subject

Figure 7. Distribution of ground truth

Point 4:

after refs. 116&117, the one modeling silicon electro-optical sensor with monolithic integration in CMOS integrated circuitry like doi: 10.1088/1361-6439/abf333 could be referred as to enhance the technical impact of the study;

Response 4:

Thank you for the suggested reference. We have included in the revised manuscript.

The discussion section is now read alike this, “This can also be developed into optical sensors-based devices and smartphone-based or webcam-based applications that might results in continuous vital signs monitoring or periodic measurement [196].” (Line 809-811)

[196] Xu, K. Silicon Electro-Optic Micro-Modulator Fabricated in Standard CMOS Technology as Components for All Silicon Monolithic Integrated Optoelectronic Systems *. J. Micromech. Microeng. 2021, 31, 054001, doi:10.1088/1361-6439/abf333.

Point 5:

after ref. 121, toward “Remote Healthcare”, the one like doi: 10.1109/ISCAS.2012.6270390 could be mentioned;

Response 5:

We have added the reference in the revised manuscript, “Though the techniques need further improvement before used in clinical medicine, home-based indoor applications can be explored already [193].” (Line 772-773)

Point 6:

at the beginning of Section “4. Discussion”, before showing 4.1, 4.2 and 4.3, one paragraph to simply introduce the connection and relationship of each sub-section should be given;

Response 6:

Thank you for your suggestion. We have included a paragraph describing the subsection's description.

Added description, “We perceive that work done in this field within the short period of this review (January 2018–April 2021) is significant and it may open up new research directions. We also believe that non-contact vital sign monitoring has gained increased attention among researchers and physicians during the current pandemic situation. This section discusses on the review’s limitations and responses to the research questions. Further, we comprehensively explain the research gaps in these fields as well as future research directions.” (Line 684-689)

Point 7:

for sub-section “4.3. Research gaps and future research directions”, some schematic flow process charts could be presented interactively with text description.

Response 7:

Thank you for your suggestion. We have added the schematic flow chart that can be found in discussion section.

Subject parameters

Camera parameters

Low-light illumination issues

Motion artifact

Real time situations

Remote health care

Research gaps & Future directions

Focus on heterogenous groups

Tuning the parameters in various environment

Signal processing and artifact removal

Requirement of robust algorithms

Smart phone applications

NIR camera might be a solution

Fusion-based algorithms

Multi-camera, multi-person, multi-vital signs

Fusion of RGB and NIR

Optical sensor based device

Figure 12. The framework of main research gaps and future directions. In this red color boxes represents the research gaps, and gray color boxes refers future directions)

Round 2

Reviewer 3 Report

Satisfactory response and revision. 

Reviewer 4 Report

No further comments